# Convex Basins in Single-Index Model Loss Landscapes: Applications to Robust Recovery under Strong Adversarial Corruption

**Santanu Das** [1] [*]   **Sagnik Chatterjee** [1] [*]   **Jatin Batra** [1]

## Abstract

We study the problem of robustly learning Gaussian Single Index Models (SIMs) in the presence of heavy-tailed noise and a constant fraction of adversarially corrupted covariates and responses. Prior work on robust recovery has considered settings such as linear regression (Pensia et al., JASA 2024), strictly monotonic link functions (Awasthi et al., NeurIPS 2022), and phase retrieval (Buna and Rebeschini, AISTATS 2025). However, these techniques do not extend to generic asymmetric non-monotonic link functions such as GELU and SWISH, which arise naturally as scalar primitives in modern gated neural architectures. We close this gap by giving the first robust recovery algorithm with near-linear sample and time complexity for generic non-monotonic link functions, thereby establishing the first robust recovery guarantees for a broad family of nonlinear SIMs for which *no guarantees were previously known*. Our central contribution is a new structural understanding of the Gaussian squared-loss landscape under adversarial contamination. Crucially, we prove that for a broad class of nonlinear non-monotonic SIMs, a dimension-independent, constant-radius convex basin exists around the ground truth and is efficiently reachable via robust spectral initialization even under adversarial contamination. Prior works fail to establish both guarantees simultaneously, thereby either breaking down under adversarial contamination or failing to handle generic non-monotonic link functions. Together, these structural insights yield a principled warm start for robust gradient descent that provably converges to a final estimation error of $O(\sigma\sqrt{\epsilon})$ in $\tilde{O}(nd)$ time with $\tilde{O}(d)$ samples, where $\epsilon$ is the contamination fraction.

[*]Equal contribution [1]School of Technology and Computer Science, Tata Institute of Fundamental Research, Mumbai, India. Correspondence to: Santanu Das <dassantanu315@gmail.com>.

*Proceedings of the 43rd International Conference on Machine Learning*, Seoul, South Korea. PMLR 306, 2026. Copyright 2026 by the author(s).

## 1. Introduction

Single-Index Models (SIMs) are a broad family of semi-parametric models subsuming linear regression, logistic regression, phase retrieval, and generalized linear models as special cases. They model the response variable $Y \in \mathbb{R}$ as a nonlinear function of a one-dimensional projection of the covariates $X \in \mathbb{R}^d$:

$$Y = f(X^\top \beta^\star) + \zeta, \qquad (1)$$

where $f : \mathbb{R} \to \mathbb{R}$ is a known *link* function, $\zeta$ is stochastic noise, and $\beta^\star \in \mathbb{R}^d$ is an unknown index vector to be recovered. The recovery of $\beta^\star$ is a fundamental problem in semi-parametric statistics (Box & Cox, 1964; McCullagh, 1984; Ichimura, 1993; Carroll et al., 1997; Hristache et al., 2001; Dalalyan et al., 2008) and machine learning (Bruna & Hsu, 2025). In this paper, we focus on Gaussian designs $X \sim \mathcal{N}(0, \mathbf{I}_d)$, the canonical setting for studying the geometry of non-convex SIM loss landscapes (Bruna & Hsu, 2025; Barbier et al., 2019; Mondelli & Montanari, 2018; Lu & Li, 2020; Arous et al., 2021; Damian et al., 2023; 2024; Joshi et al., 2025).

While classical SIM recovery techniques relied on clean-data assumptions (Härdle & Stoker, 1989; Li, 1991), in practice, data is invariably subject to noise and corruption. The field of robust statistics (Huber, 1992; Tukey, 1975; Hampel et al., 22 March 2005) developed estimators with well-understood robustness guarantees such as high breakdown points, primarily in low-dimensional settings. However, in high dimensions, achieving strong robustness often leads to estimators that are computationally intractable, with many classical formulations known to be NP-hard (Johnson & Preparata, 1978; Bernholt, 2006). Recent breakthroughs in algorithmic robust statistics (Diakonikolas et al., 2019a;c) overcame this computational barrier by providing efficient high-dimensional subroutines for robust mean estimation and robust PCA that tolerate both heavy-tailed noise and strong adversarial corruption. These subroutines serve as algorithmic building blocks for efficient robust recovery in structured models such as linear regression (Cherapanamjeri et al., 2020; Prasad et al., 2020), logistic regression (Diakonikolas et al., 2019b; 2022), and phase retrieval (Dong et al., 2025; Buna & Rebeschini, 2025; Das & Batra, 2026).

For monotonic link functions, Kalai & Sastry (2009) gave the first efficient recovery algorithm for monotone Lipschitz SIMs in the clean setting via the Isotron algorithm, later extended to broader classes of SIMs by Kakade et al. (2011) and Plan & Vershynin (2016). Under heavy-tailed noise and strong adversarial contamination, near-linear time algorithms with optimal sample complexity $\tilde{O}(d)$ were obtained for linear regression, i.e., $f(x) = x$, (Cherapanamjeri et al., 2020; Prasad et al., 2020) with Pensia et al. (2025) later obtaining information-theoretically optimal error rates under the same guarantees. Beyond linear regression, monotonic link functions such as logistic regression have been studied in the context of Generalized Linear Models (GLMs) (Awasthi et al., 2022); for robust recovery under heavy-tailed noise and strong adversarial contamination, Diakonikolas et al. (2019b) proposed a polynomial-time, optimal-sample-complexity algorithm, while Diakonikolas et al. (2022) later obtained near-linear time at the cost of polynomial sample complexity in the streaming setup. For non-monotonic link functions, phase retrieval, i.e., $f(z) = z^2$, stands out as the canonical example, with applications in optics, crystallography, X-ray imaging, and astrophysics. Candes et al. (2015) and Netrapalli et al. (2015) gave the first efficient recovery algorithms under Gaussian covariates in the absence of noise and corruption. Under strong adversarial contamination without noise ($\zeta = 0$), Dong et al. (2025) proposed a near-linear time algorithm with optimal sample complexity $\tilde{O}(d)$. Buna & Rebeschini (2025) extended these results to heavy-tailed noise with an exponential-time algorithm, which was recently subsumed by Das & Batra (2026), who gave a polynomial-time optimal sample complexity robust recovery algorithm for the same setting.

We note that the lack of robust recovery results for generic non-monotonic SIMs beyond phase retrieval is telling: standard first-order proof techniques (Arous et al., 2021; Ren et al., 2025; Arous et al., 2025) break down under strong adversarial contamination in high dimensions, since the adversary destroys the statistical structure that gradient-based convergence arguments rely upon. The special case of phase retrieval enjoys two structural properties that are conducive for robust recovery: (i) its squared-loss landscape admits a convex basin of constant radius around the true parameter, enabling second-order convergence guarantees within the basin, and (ii) its non-vanishing second Hermite coefficient[1] ensures the signal direction is reachable via spectral methods (Candes et al., 2015; Netrapalli et al., 2015). It is a priori unclear whether these properties extend to generic non-monotonic SIMs, which often possess a high information exponent[2] (IE). Even in the clean setting, assuming one

existed, accessing a convex basin for generic SIMs would require computationally demanding methods such as Tensor PCA (Anandkumar et al., 2017), in stark contrast to the simple spectral methods that suffice for phase retrieval. Indeed, it remains unclear for which classes of non-monotonic link functions efficient provable robust recovery is achievable. This uncertainty naturally brings us to the following question:

**Question 1.** Can we characterize the class of link functions which admit *efficient* provable robust recovery guarantees under heavy-tailed noise and strong adversarial contamination?

**Our Contributions:** We make significant progress on the above question by identifying two general structural conditions on the link function $f$ outlined in Assumptions 2.1 and 2.2, under which efficient provable robust recovery is achievable. These conditions are remarkably general, capturing a large class of SIMs with generative exponent[3] at most 2. This class includes phase retrieval, TANH, PROBIT, and LOGISTIC, as well as modern activation functions such as GELU (Hendrycks & Gimpel, 2023), SWISH (Ramachandran et al., 2018) which arise naturally as scalar primitives in gated neural architectures (Shazeer, 2020) that serve as the fundamental building blocks of Transformer architectures (e.g., GPT and BERT (Radford et al., 2018; Devlin et al., 2019)). We now state an informal characterization of our main result below.

**Theorem 1.1** (Linear Sample and Time Robust Recovery, Informal). *Consider a SIM with a link function satisfying Assumptions 2.1 and 2.2. There exists an algorithm that, using $n = \tilde{O}(d)$ samples and tolerating a constant fraction $\epsilon$ of adversarial contamination under heavy-tailed noise with variance $\sigma^2$, outputs an estimate $\hat{\beta}$ s.t. $\|\hat{\beta} - \beta^\star\|_2 = O(\sigma\sqrt{\epsilon})$ in $\tilde{O}(nd)$ time, with high probability.*

### 1.1. Technical Overview

A natural strategy for robust recovery in generic non-monotonic SIMs is to leverage the geometry of the squared-loss landscape. Global convexity of the loss landscape would guarantee that any optimizer avoids spurious local minima, but for generic non-monotonic SIMs, global convexity of the squared loss holds if and only if $f$ is affine, reducing to linear regression. For non-affine link functions, a straightforward approach is therefore to establish the existence of a convex neighborhood around the true parameter $\beta^\star$. Once an optimizer enters this region, standard convex

---

[1]The $k$-th Hermite coefficient of $f$ is the coefficient of the $k$-th Hermite polynomial in the Gaussian expansion of $f$.

[2]The information exponent (IE) of a link function is the order

of its first non-zero Hermite coefficient; a high IE means the signal is suppressed in low-order moments.

[3]The generative exponent (GE) of a link function is the minimum IE achievable over all square-integrable label transforms of $f$ (Damian et al., 2024). For SIMs having GE at most 2, the signal is detectable via second-order spectral methods after an appropriate label transform.

optimization guarantees ensure reliable recovery. However, for this strategy to be computationally viable in high dimensions, the convex basin must have radius $R = O(1)$ strictly independent of the ambient dimension $d$, since a vanishing basin of attraction offers no algorithmic guarantee in high-dimensional non-convex optimization. Prior to this work, such a dimension-independent convex basin was known to exist only for phase retrieval and monotonic link functions. This motivates our first contribution.

**Contribution 1.** We identify a sufficient condition (see Assumption 2.1) under which the squared-loss landscape admits a convex basin of dimension-independent, constant radius $R = O(1)$ around the true parameter $\beta^\star$, for a wide class of link functions including GELU, SWISH, TANH, PROBIT, LOGISTIC, and phase retrieval. This is the first such guarantee for any non-affine, non-monotonic link function beyond phase retrieval.

Establishing the existence of a convex basin does not by itself yield a computational guarantee. One must also certify that the basin can be reached efficiently from an initialization, even under heavy-tailed noise and strong adversarial contamination. The recent works of Buna & Rebeschini (2025) and Das & Batra (2026) make progress in this direction for phase retrieval, but their approaches suffer from two limitations. First, the robust PCA subroutines they employ run in time polynomial in the dimension. Second, and more fundamentally, their reachability arguments exploit the symmetric structure of the quadratic link and do not extend to asymmetric non-monotonic link functions such as GELU and SWISH. Beyond these computational limitations, certifying reachability under strong adversarial contamination in high dimensions poses a deeper analytical challenge. Prior proof techniques (Arous et al., 2021; Ren et al., 2025; Arous et al., 2025) that implicitly leverage convex basin structure rely on martingale-drift decompositions that require the stochastic deviations to be mean-zero, a property the adversary destroys by corrupting a constant fraction of samples in high dimensions. Our second contribution resolves both the computational and analytical obstacles.

**Contribution 2.** We identify a sufficient condition (see Assumption 2.2), termed *Expected Squared Convexity (ESC)*, which characterizes when the leading eigenvector of (higher-order) moment estimators aligns with the signal $\beta^\star$. Further, under the ESC condition, for link functions whose squared-loss landscape admits a constant-radius convex basin around $\beta^\star$, off-the-shelf robust spectral initialization on these (higher-order) sample moment matrices yields an estimate $\beta_0$ that (i) lies in the convex basin under heavy-tailed noise and strong adversarial contamination, and (ii) estimates the true parameter $\beta^\star$ with additive error $O(\epsilon^{1/4})$. This is the first explicit, efficiently checkable guarantee that a constant-radius convex basin is reachable for generic non-monotonic SIMs under adversarial contamination.

Contribution 2 demonstrates that the shortcomings of the previous approaches (Buna & Rebeschini, 2025; Das & Batra, 2026) are not intrinsic. By coupling second-order Stein identities with a refined analysis of robust spectral estimation, we generalize the reachability insights of phase retrieval to a broad class of non-monotonic link functions while achieving near-linear initialization time. Under the Gaussian design, second-order Stein's identity allows us to decompose the population moment matrix into a rank-one component aligned with $\beta^\star$ and an isotropic component. Crucially, we show that the coefficient of the signal direction in this decomposition is governed precisely by the ESC condition, which allows us to identify the moment matrix whose leading eigenvector is $\beta^\star$. Using this structural insight, we apply the off-the-shelf robust PCA subroutine of Jambulapati et al. (2024), which explicitly leverages the hypercontractivity of the Gaussian design to extract the leading eigenvector of the sample moment matrices efficiently under adversarial contamination. This contrasts with earlier analyses (Buna & Rebeschini, 2025; Das & Batra, 2026; Yang et al., 2017), which either fell short of this general structural characterization or did not exploit such concentration properties to ensure both robustness and efficiency.

However, attaining better error rates requires going beyond spectral initialization. In classical phase retrieval algorithms like Wirtinger Flow (Candes et al., 2015), spectral methods are used primarily to initialize the optimization within a basin of attraction, after which Gradient Descent (GD) is employed to converge to the exact solution. In the robust setting, this two-stage architecture is equally critical but for a fundamental statistical reason: we observe that relying solely on robust spectral estimators encounters a fundamental error floor of $O(\epsilon^{1/4})$. To break this barrier, we use an off-the-shelf robust Gradient Descent (GD) subroutine that refines the initial estimate significantly while maintaining near-linear time complexity.

**Contribution 3.** We provide the *first* near-linear time, *optimal* sample complexity algorithm for robust recovery of a wide class of link functions under heavy-tailed noise and strong adversarial contamination. By initializing via higher-order robust spectral methods and optimizing with robust GD, we achieve a significantly better additive estimation error of $O(\sqrt{\epsilon})$. Notably, this constitutes the first efficient robust recovery guarantee for any non-monotonic link function beyond phase retrieval, in particular for widely used activations such as GeLU and Swish.

### 1.1.1. RELATED WORK

To put our contributions in context, prior to this work, it was not known if *efficient* robust recovery in under both heavy-tailed noise and strong adversarial contamination was possible for generic link functions. We now give a brief overview of known results for robust recovery in SIMs

beyond linear regression, and defer a detailed literature review to the appendix (see Section A).

Diakonikolas et al. (2019b) proposed a polynomial-time $\tilde{O}(d)$ sample recovery algorithm (SEVER) for logistic regression under strong adversarial contamination with respect to the hinge loss and the logistic loss. With respect to the squared loss, their sample complexity blows up to $\tilde{O}(d^5)$. Diakonikolas et al. (2022) obtain a near-linear time robust recovery streaming algorithm for logistic regression that requires $\tilde{O}(d^2)$ samples, with respect to the squared loss. We note that both Diakonikolas et al. (2019b) and Diakonikolas et al. (2022) obtain an error of $\sigma\sqrt{\epsilon}$, similar to us. Awasthi et al. (2022) obtain optimal sample complexity robust recovery guarantees under strong adversarial contamination for GLMs (which rely on only monotonic link functions) *without any guarantees* on the running time of their approach, and only under Gaussian noise, in contrast to our near-linear time robust recovery algorithm under heavy-tailed noise and strong adversarial contamination. However, our error rate of $O(\sigma\sqrt{\epsilon})$ is worse than their $O(\sigma\epsilon \log \frac{1}{\epsilon})$ error rate, which is *optimal*. The caveat, however, is that our error rate also holds for a large class of non-monotonic link functions, which are not addressed by Awasthi et al. (2022). We also remark that our error rate matches the best known error rate for existing robust recovery algorithms for non-monotonic link functions (Diakonikolas et al., 2019b; 2022; Buna & Rebeschini, 2025; Das & Batra, 2026) under heavy-tailed noise and strong adversarial contamination.

### 1.2. Organization of the paper

We detail our problem setup, model, and technical tools in Section 2. In Section 3, we prove the existence of a convex basin. Next, we obtain a linear-time optimal-sample complexity robust recovery algorithm in Section 4. Finally in Section 5, we discuss our contributions and outline multiple avenues for future directions.

## 2. Preliminaries

**Notation** For any positive integer $n$, let $[n]$ denote the set $\{1, 2, \ldots, n\}$. For a vector $v \in \mathbb{R}^d$, $\|v\|_2$ denotes its Euclidean norm. For two vectors $u, v \in \mathbb{R}^d$, $\langle u, v \rangle$ denotes their inner product. For a matrix $M$, $\|M\|_{\text{op}}$ denotes its operator norm, $\text{Tr}(M)$ its trace, and we write $M \succeq 0$ to indicate that $M$ is PSD. We denote the identity matrix in $d$ dimensions by $\mathbf{I}_d$. The indicator function of an event $E$ is denoted by $\mathbb{I}(E)$. $\mathcal{N}(\mu, \Sigma)$ denotes a multivariate Gaussian with mean $\mu$ and covariance $\Sigma$. $\mathbb{E}[\cdot]$ denotes expectation. $\tilde{O}(\cdot)$ hides logarithmic factors in the dimension $d$, sample size $n$, corruption level $\epsilon$, and failure probability $\delta$. Finally, $f'$ and $f''$ denote the first and second derivatives of the link function $f$, respectively.

**Problem Setup** We consider the single index model (SIM) defined in (1), with covariates $X \in \mathbb{R}^d$ and a known[4] link function $f$. In this section, we formally introduce the problem our paper tackles. We first define the notion of strong adversarial contamination, and then introduce our model.

**Definition 2.1** (Strong Adversarial Contamination (Diakonikolas et al., 2019a))**.** Given a contamination tolerance level $\epsilon \in (0, 1/2)$ and a distribution $\mathcal{D}$ on $\mathbb{R}^d$, generate a clean training set by drawing $n$ samples from $\mathcal{D}$. The adversary is allowed to inspect the entire training set and alter up to $\epsilon n$ of the clean samples arbitrarily. This modified training set of $n$ points is then provided as input to the algorithm. We refer to the modified training set as $\epsilon$-strongly contaminated dataset.

**Definition 2.2** (The Model)**.** A dataset of the form $\{(x_i, y_i)\}_{i=1}^n$ is generated such that the responses $y_i$ are drawn from an unknown parameter $\beta^\star \in \mathbb{R}^d$ according to Equation (1), with covariates $x_i \sim \mathcal{N}(0, \mathbf{I}_d)$, a link function $f$ satisfying Assumptions 2.1 and 2.2, and $\|\beta^\star\|_2 = 1$.[5] The noise variables $\zeta_i$ are heavy-tailed, zero-mean conditioned on $x_i$, and homoscedastic with bounded variance and fourth moments, i.e., $\mathbb{E}[\zeta_i \mid x_i] = 0$, $\mathbb{E}[\zeta_i^2 \mid x_i] = \sigma^2$, and $\mathbb{E}[\zeta_i^4 \mid x_i] = K_4^4$. An $\epsilon$ fraction of the dataset $\{(y_i, x_i)\}_{i=1}^n$ is then corrupted by a strong adversary, as in Definition 2.1.

**Definition 2.3** (The Robust Recovery Problem for SIMs)**.** Given access to an $\epsilon$-strongly contaminated training set $\{(y_i, x_i)\}_{i=1}^n$ (as described in Definition 2.2), output w.h.p., a unit norm estimate $\beta \in \mathbb{R}^d$ s.t. $\|\beta - \beta^\star\|_2 = O(\sigma\sqrt{\epsilon})$.

We now state the natural structural assumptions we require on the link function. The first assumption controls the smoothness of the population loss landscape through one-dimensional expectations of $f$ and its derivatives under the Gaussian measure.

**Assumption 2.1.** Given a link function $f$, there exists a radius $R > 0$ satisfying

$$R \leq \frac{\mu}{(2(315)^{1/4}C_{\text{lip}}(R))},$$

where $C_{\text{lip}}(R)$ is defined as

$$C_{\text{lip}}(R) := \sup_{\|\beta - \beta^\star\| \leq R} \sqrt{\mathbb{E}_{z \sim \mathcal{N}(0, \|\beta\|^2)}[g(z)]},$$

and $g(z) := 18f'(z)^2 f''(z)^2 + 2f'''(z)^2 f(z)^2$.

In Assumption 2.1, the local Lipschitz complexity measure $C_{\text{lip}}(R)$ measures how quickly the curvature of the population loss can deteriorate as $\beta$ moves away from $\beta^\star$. A

---

[4]In non-parametric setups, $f$ is assumed unknown (Zhu & Xue, 2006), whereas in machine learning it is either known or constrained to a well-behaved class (Kalai & Sastry, 2009).

[5]Although this is standard practice (Bruna et al., 2023; Yang et al., 2017; Dong et al., 2025), we leave the removal of this assumption to future work.

simple upper bound on $C_{\mathrm{lip}}(R)$ (via Cauchy–Schwarz) is $5(M_2 M_3 + M_1 M_4)$, where

$$M_k := \sup_{\|\beta - \beta^\star\| \le R} \left( \mathop{\mathbb{E}}_{Z \sim \mathcal{N}(0, \|\beta\|_2^2)} \left[ f^{(k-1)}(Z)^4 \right] \right)^{\frac{1}{4}}, k \in [4],$$

where $f^{(0)} := f$, $f^{(1)} := f'$, $f^{(2)} := f''$, $f^{(3)} := f'''$. Crucially, since $g$ is evaluated pointwise on $Z \sim \mathcal{N}(0, \|\beta\|_2^2)$, the quantities $M_1, M_2, M_3, M_4$ (and hence $C_{\mathrm{lip}}(R)$) are determined entirely by one-dimensional Gaussian integrals of $f$ and its derivatives. Therefore, any link function satisfying Assumption 2.1 has a dimension-independent basin radius $R$.

*Remark* 2.1. We note that Assumption 2.1 only imposes a mild regularity condition on the link function $f$. We only require $f$ and its first three derivatives $f', f'', f'''$ to have finite fourth moments under the Gaussian measure. This condition is satisfied by all activation functions with at most polynomial growth. On the flip side, link functions that violate this condition necessarily grow faster than any polynomial, which implies $\mathbb{E}[f(X^\top \beta^\star)^2]$ is not bounded even under Gaussian covariates!

The second assumption complements the first by addressing a different question: while Assumption 2.1 guarantees that the loss landscape admits a dimension-independent convex basin near $\beta^\star$, it does not guarantee (efficient) reachability to this convex basin from a random initialization. Assumption 2.2, termed Expected Squared Convexity (ESC), precisely characterizes when the signal direction $\beta^\star$ is identifiable from the second-order moments of the data, enabling a robust spectral initialization that lands within the convex basin.

**Assumption 2.2.** Let $f$ be a twice differentiable link function, and let $X \sim \mathcal{N}(0, \mathbf{I}_d)$. For any $\beta \in \mathbb{R}^d$, the expected squared convexity of the link function $f : \mathbb{R} \to \mathbb{R}$ at $\beta$ is defined as

$$\mathrm{ESC}(\beta, f) := \mathbb{E}\left[ \left( f'(X^\top \beta) \right)^2 + f(X^\top \beta)\, f''(X^\top \beta) \right].$$

A twice differentiable link function $f$ is strictly ESC if $\mathrm{ESC}(\beta^\star, f) > 0$.

*Remark* 2.2. To see why ESC is the right condition for identifiability in non-monotonic links, note that by the product rule, $\mathrm{ESC}(\beta, f) = \mathbb{E}\left[ \left( f^2(X^\top \beta) \right)'' \right]$. Hence, ESC is a higher-order analogue of monotonicity: just as $\mathbb{E}[f'(Z)] > 0$ ensures that the first-order moments of the data carry information about $\beta^\star$, $\mathrm{ESC}(\beta^\star, f) > 0$ ensures that the second-order moments of the data carry information about $\beta^\star$.

## 2.1. Tools

In this subsection, we present the main tools used in our work.

**Lemma 2.1** (Second-order Multivariate Stein's Lemma). *Let* $X \sim \mathcal{N}(0, \mathbf{I}_d)$. *For any twice-differentiable function* $g : \mathbb{R}^d \mapsto \mathbb{R}$ *s.t.* $\mathbb{E}\left[\nabla^2 g(X)\right]$ *exists, we have* $\mathbb{E}\left[g(X)(XX^T - \mathbf{I}_d)\right] = \mathbb{E}\left[\nabla_X^2\, g(X)\right].$

The univariate version of Lemma 2.1 states that for a Gaussian variable $z \sim \mathcal{N}(0, 1)$, $\mathbb{E}[g(z)(z^2 - 1)] = \mathbb{E}[g''(z)]$. We now present the main results from algorithmic robust statistics that our paper builds upon.

**Lemma 2.2** (Robust Mean Estimation(Theorem 3.2 of Diakonikolas et al. (2022))). *Let* $\mathcal{D}$ *be a distribution on* $\mathbb{R}^d$ *with unknown mean* $\mu$ *and covariance* $\Sigma$ *satisfying* $\Sigma \preceq \sigma'^2 I$, *for some constant* $\sigma' > 0$. *For* $\epsilon$ *smaller than a sufficiently small universal constant and* $\delta > 0$, *given an* $\epsilon$-corrupted dataset (see Definition 2.1) of $n = \tilde{O}(d/\epsilon)$ *samples, there exists an algorithm running in time* $\tilde{O}(nd\,\mathrm{polylog}(d, n, 1/\epsilon, 1/\delta))$ *that outputs* $\widehat{\mu}$ *satisfying, w.p.* $\ge 1 - \delta$, $\|\widehat{\mu} - \mu\|_2 = O(\sigma' \sqrt{\epsilon})$.

We use a robust PCA subroutine of (Jambulapati et al., 2024), for which we recall two definitions. For $M \in \mathbb{S}_{\succeq 0}^{d \times d}$ and $\rho \in [0, 1]$, a unit vector $u \in \mathbb{R}^d$ is a $\rho$-approximate energy 1-PCA of $M$ (Jambulapati et al., 2024, Definition 2) if $\langle uu^\top, M \rangle \ge (1 - \rho)\|M\|_1$, where $\|M\|_1 := \max_{\|v\|_2 = 1} \langle vv^\top, M \rangle$. A random vector $X \in \mathbb{R}^d$ is $(p, C_p)$-hypercontractive (Jambulapati et al., 2024, Definition 8) if for all $u \in \mathbb{R}^d$, $\mathbb{E}[\langle u, X \rangle^p]^{1/p} \le C_p (\mathbb{E}[\langle u, X \rangle^2])^{1/2}$, for some constant $C_p$ and even integer $p$. We now state the linear-time robust PCA guarantee (Jambulapati et al., 2024, Theorem 4).

**Lemma 2.3** (Robust hypercontractive 1-ePCA). *Let* $\mathcal{D}$ *be a* $(4, C_4)$-*hypercontractive distribution on* $\mathbb{R}^d$ *with second moment matrix being* $\Sigma$.[6] *Let* $\epsilon \in (0, \epsilon_0)$, $\delta \in (0, 1)$, *and* $\rho = \Theta(C_4^2 \sqrt{\epsilon}) \in (0, \rho_0)$ *for absolute constants* $\epsilon_0, \rho_0 > 0$. *Given an* $\epsilon$-corrupted dataset $T$ *of size* $|T| = \Theta\left( \vartheta \cdot \frac{d \log d + \log(1/\delta)}{\rho^2} \right)$, *where* $\vartheta := C_4^6/\sqrt{\epsilon}$, *there exists an algorithm* $\mathcal{A}_k$,[7] *that takes* $T, \epsilon, \rho, \delta$ *as input and outputs a unit vector* $\hat{u} \in \mathbb{R}^d$ *in time* $O\left( \frac{nd}{\rho^2} \mathrm{polylog}\left( \frac{d}{\epsilon \delta} \right) \right)$ *such that, w.p.* $\ge 1 - \delta$, $\hat{u}$ *is an* $O(\rho)$-*approximate energy* 1-*PCA of* $\Sigma$.

## 3. Existence of Convex Basins in the Loss Landscape of SIMs

Consider the population loss $\mathcal{L}(\beta) := \frac{1}{2}\mathbb{E}[(f(X^\top \beta) - Y)^2]$ with the corresponding Hessian $H(\beta) := \mathbb{E}[(f'(X_i^\top \beta))^2 + (f(X_i^\top \beta) - Y_i)f''(X_i^\top \beta)) X_i X_i^\top]$. We now give a sufficient condition to characterize a class

---

[6]Jambulapati et al. (2024) provide robust PCA guarantees for covariance matrices; these results apply equally to second-moment matrices.

[7]$\mathcal{A}_k$ refers to Algorithm 1 of (Jambulapati et al., 2024) with Algorithm 2 of (Diakonikolas et al., 2023) as the 1-ePCA oracle.

of link functions that admit a dimension-independent constant-sized convex basin around $\beta^\star$ in the loss landscape, in the following theorem:

**Theorem 3.1.** *Let* $Z \sim \mathcal{N}(0,1)$. *Define the second and fourth moment proxies,* $\mu = \min\{\mathbb{E}[f'(Z)^2], \mathbb{E}[Z^2 f'(Z)^2]\}$, *and* $\mu_1 = \max\{\mathbb{E}[f'(Z)^2], \mathbb{E}[Z^2 f'(Z)^2]\}$. *Consider the model as given in Definition 2.2, s.t. the link function* $f$ *satisfies Assumption 2.1 with radius* $R > 0$, *then for all* $\beta$ *in the Euclidean ball* $\mathcal{B}(\beta^\star, R)$, *the Hessian* $H(\beta) := \nabla^2 \mathcal{L}(\beta)$ *satisfies*

$$\frac{\mu}{2} \mathbf{I}_d \preceq H(\beta) \preceq \left(\frac{\mu}{2} + \mu_1\right) \mathbf{I}_d.$$

Theorem 3.1 establishes that whenever the link function satisfies Assumption 2.1, the population loss landscape is $\frac{\mu}{2}$-strongly convex and $\frac{\mu+2\mu_1}{2}$-smooth throughout $\mathcal{B}(\beta^\star, R)$. Crucially, both the strong convexity constant $\mu/2$ and the basin radius $R$ are *dimension-independent*: they are determined entirely by one-dimensional integrals of $f$ and its derivatives against the standard Gaussian. This dimension-independence is the key structural fact that enables our robust recovery guarantees in Section 4, since it allows the curvature of the loss to dominate the adversarial bias uniformly over the entire ball $\mathcal{B}(\beta^\star, R)$, regardless of the ambient dimension $d$. We present an extended proof sketch below and defer the full proof to Section B in the appendix.

*Proof Sketch of Theorem 3.1.* The proof proceeds in three stages: (i) computing the exact spectral structure of the population Hessian at $\beta^\star$; (ii) bounding the operator-norm deviation of the empirical Hessian as $\beta$ moves away from $\beta^\star$; and (iii) combining these to certify strong convexity over the full ball $\mathcal{B}(\beta^\star, R)$.

**Hessian Decomposition.** We decompose the Hessian at an arbitrary $\beta$ near $\beta^\star$ as $H(\beta) = H(\beta^\star) + \Delta(\beta)$, where $\Delta(\beta) := H(\beta) - H(\beta^\star)$ represents the deviation of the curvature due to the nonlinearity of the link function $f$. To ensure local strong convexity, we require $\lambda_{\min}(H(\beta)) > 0$. By Weyl's inequality, we know that $\lambda_{\min}(H(\beta)) \geq \lambda_{\min}(H(\beta^\star)) - \|\Delta(\beta)\|_{\mathrm{op}}$.

**Strong Convexity at Optimum.** The Hessian $H(\beta^\star)$ at the population solution decomposes into a curvature term $\mathbb{E}[(f'(X^\top \beta^\star))^2 XX^\top]$ and a residual term $\mathbb{E}[(f(X^\top \beta^\star) - Y)f''(X^\top \beta^\star)XX^\top]$. The residual term vanishes because $\mathbb{E}[Y|X] = f(X^\top \beta^\star)$. For the curvature term, we exploit Gaussian symmetry (see Lemma B.1) to obtain the explicit closed-form expression of $\mathbb{E}[(f'(Z))^2 XX^\top]$ as

$$\mathbb{E}[(f'(Z))^2]\mathbf{I}_d + (\mathbb{E}[Z^2(f'(Z))^2] - \mathbb{E}[(f'(Z))^2])\beta^\star\beta^{\star\top}.$$

This matrix has two distinct eigenvalues: $\mathbb{E}[Z^2 f'(Z)^2]$

corresponding to the eigenvector $\beta^\star$, and $\mathbb{E}[f'(Z)^2]$ corresponding to all directions orthogonal to $\beta^\star$. Thus, $\mu := \lambda_{\min}(H(\beta^\star)) = \min\{\mathbb{E}[f'(Z)^2], \mathbb{E}[Z^2 f'(Z)^2]\}$.

**Perturbation Analysis:** To extend this convexity to a ball of radius $R$, we bound the operator norm of the perturbation $\Delta(\beta)$. We recall that the Hessian at a general $\beta$ can be written as $H(\beta) = \mathbb{E}[Q(X^\top \beta, X^\top \beta^\star)XX^\top]$, where $Q(z, z^*) = (f'(z))^2 + f''(z)(f(z) - f(z^*))$. The entry-wise difference is thus driven by $\Delta Q(z, z^*) = Q(z, z^*) - Q(z^*, z^*)$, where $z = X^\top \beta$ and $z^* = X^\top \beta^\star$. Applying the Mean Value Theorem to $Q$ along the path between $\beta$ and $\beta^\star$, we define an auxiliary function $A(z, z^*)$ involving up to the third derivative of $f$. The spectral norm is bounded via the Cauchy-Schwarz inequality and higher-order moment bounds of the Gaussian:

$$\|\Delta(\beta)\|_{\mathrm{op}} = \sup_{v:\|v\|=1} |v^\top \mathbb{E}[\Delta Q \cdot XX^\top]v|$$
$$\leq (365)^{1/4} \cdot \sqrt{\mathbb{E}[A(z, z^*)^2]} \cdot \|\beta - \beta^\star\|_2,$$

where $(365)^{1/4}$ is a constant derived from the higher-order moments of the standard normal.

**Establishing the Convex Basin:** A critical step in the proof is bounding the term $\mathbb{E}[A(z, z^*)^2]$. The function $A$ evaluates derivatives of $f$ at interpolated points $\lambda z + (1-\lambda)z^*$, where $z = X^\top \beta$, and $z^* = X^\top \beta^\star$. Hence, any point on this path is a linear projection of the Gaussian vector $X$. Because $X \sim \mathcal{N}(0, \mathbf{I}_d)$, the dot product of $X$ with any fixed vector $u \in \mathbb{R}^d$ is distributed as $\mathcal{N}(0, \|u\|^2)$. Thus, the r.v. representing the interpolation point is distributed as $Z \sim \mathcal{N}(0, \sigma^2)$, where $\sigma^2 = \|\lambda\beta + (1-\lambda)\beta^\star\|^2$ depends only on the length of the interpolation vector. We define the Lipschitz constant $C_{\mathrm{lip}}(R)$ by taking the supremum over the entire Euclidean ball $\mathcal{B}(\beta^\star, R)$. Crucially, because $C_{\mathrm{lip}}(R)$ is defined entirely by 1D integrals of the link function, it is **independent of the ambient dimension** $d$. By choosing the radius $R < \frac{\mu}{2 \cdot (365)^{1/4} \cdot C_{\mathrm{lip}}(R)}$, we have $\|\Delta(\beta)\|_{\mathrm{op}} \leq \mu/2$, implying $\lambda_{\min}(H(\beta)) \geq \mu/2$. $\square$

## 4. Linear-Time Robust Recovery of SIMs

In this section, we present our linear-sample and time algorithm for robustly recovering the true signal $\beta^\star$ when the link function admits a convex basin in the loss landscape under heavy-tailed noise and strong adversarial contamination. We now formally state our main result.

**Theorem 4.1** (Linear-time Algorithm for Robust Recovery). *Consider the model in Definition 2.2. Define* $C_{\mathrm{lip}}(R)$ *and* $R$ *as in Theorem 3.1. Define* $\alpha = \frac{\mu}{2} + \mu_1, \gamma = \frac{\mu}{2}$ *denote the smoothness and strong convexity parameters of Theorem 3.1. Define*

$$\phi_1 := \sup_{\beta \in \mathcal{B}(\beta^\star, R)} \mathbb{E}[f'(X^\top \beta)^{16}]^{1/4},$$

*and*

$$\phi_2 := \sup_{\beta \in \mathcal{B}(\beta^\star, R)} \mathbb{E}\big[f'(X^\top \beta)^4\big]^{1/2},$$

*and assume $K_4 \leq K$. Define $c := \mathrm{ESC}(\beta^\star; f)$, and let $C_4$ be the hypercontractivity parameter as defined in Lemma 4.2. Algorithm 1 takes $n = \tilde{O}(m + P\tilde{m})$ samples from an $\epsilon$-contaminated dataset $T$, such that*

$$\epsilon = O\left(\min\left\{\frac{1}{C_4^4}, \frac{c^2 \min\{R^4, 1\}}{C_4^4 \left(\sigma^2 + \mathbb{E}[f^2] + c\right)^2}, \frac{\gamma^2}{\phi_1}, \frac{\gamma^2 R^2}{\sigma^2 \phi_2}\right\}\right),$$

*and outputs an estimate $\beta$ of the true parameter in time $\tilde{o}\left(\frac{md}{C_4^4} + P\tilde{m}d\right)$ w.h.p., s.t.*

$$\|\beta - \beta^\star\| = O(\sigma\sqrt{\epsilon}), \tag{2}$$

*where $m = \Theta\left(C_4^2 \cdot \left(\frac{d \log d + \log(1/\delta)}{\epsilon^{3/2}}\right)\right), \tilde{m} = \tilde{O}(d/\epsilon)$ and $P = O(1)$ denotes the number of iterations of the LRGD algorithm (see Algorithm 3).*

In the above Theorem 4.1, $m$ is the sample complexity for the robust spectral initialization subroutine (see Algorithm 2) and $\tilde{m}$ is the sample complexity for the robust gradient descent subroutine (see Algorithm 3). We begin by giving the pseudocode of Algorithm 1 in Theorem 4.1 along with a high-level overview.

---

**Algorithm 1** Linear-time Algorithm for Robust Recovery

---

1: **Input:** Samples $S = \{(x_i, y_i)\}_{i=1}^N$, Corruption $\epsilon$, parameters $P, \alpha, \gamma$.
2: Randomly partition the $N$ samples into $P + 1$ disjoint buckets of equal sizes, denoted by $N_1, N_2, \ldots, N_{P+1}$.
3: $\beta_0 \leftarrow \mathrm{LRSI}(N_1, \epsilon)$      # Initialize in convex basin
4: $\beta_P \leftarrow \mathrm{LRGD}(N_2 \ldots N_{P+1}, \beta_0, \epsilon, \alpha, \gamma)$
5: **Output:** $\beta_P / \|\beta_P\|$.

---

First, we begin by recalling that if our link function $f$ satisfies Assumptions 2.1 and 2.2, the population loss-landscape admits a convex basin in a neighborhood of $\beta^\star$ (see Theorem 3.1). Algorithm 1 exploits this structure via the LRSI subroutine (see Algorithm 2) which combines generalized higher-order Stein's identities (see Lemma 2.1) together with the linear-time robust hypercontractive 1-ePCA algorithm as described in Lemma 2.3 to construct an estimate $\beta_0$ of the true signal $\beta^\star$ that lies within the convex basin. Finally, using $\beta_0$ as a warm start, the LRGD algorithm (see Algorithm 3) converges to $\beta^\star$. We first carefully detail the robust spectral initialization step, then the robust gradient descent step, and finally the proof of Theorem 4.1.

### 4.1. Warm Start via Linear time Spectral Initialization

**Lemma 4.1.** *Consider the model given in Definition 2.2 and define $\tilde{Y} := YX$. Then, $\beta^\star$ is the top eigenvector of*

$\mathbb{E}\left[\tilde{Y}\tilde{Y}^T\right]$ *with eigenvalue* $\lambda_{\max} = \sigma^2 + \mathbb{E}\left[(f(X^T\beta^\star))^2\right] + 2\mathbb{E}\left[(f'X^T\beta^\star)^2 + f(X^T\beta^\star) \cdot f''(X^T\beta^\star)\right]$.

Lemma 4.1 and Assumption 2.2 state that for the random variable $\tilde{Y} = YX$, where $Y, X$ are defined in Definition 2.2 s.t. the link function $f$ satisfies Assumption 2.2, the vector $\beta^\star$ is the leading eigenvector of the matrix $\mathbb{E}[\tilde{Y}\tilde{Y}^\top]$, with eigenvalue $\lambda_{\max}$. For a warm start, we robustly estimate the leading eigenvector of the second-moment matrix of $\tilde{Y}$. To this end, we employ the robust hypercontractive 1-ePCA algorithm. The applicability of this method requires the distribution of $\tilde{Y}$ to be hypercontractive, a property we establish in Lemma 4.2.

**Lemma 4.2.** *Consider the model in Definition 2.2. Then, $\tilde{Y} = YX$ is $(4, C_4)$ hypercontractive, where*

$$C_4 = 3\left(\mathbb{E}\left[f\left(X^\top\beta^\star\right)^8\right]^{1/8} + K_4\right)/\sigma.$$

We now state the guarantees for the LRSI algorithm (see Algorithm 2) that uses the PCA algorithm of (Jambulapati et al., 2024) as a subroutine.

---

**Algorithm 2** Linear-Robust-Spectral-Initialization (LRSI)

---

1: **Input:** Sample sets $N_1$, corruption level $\epsilon$.
2: **Output:** Initial estimate $\beta_0$
3: Consider the samples $N_1$, define $X'_j = y_j x_j$. Apply the Robust hypercontractive 1-ePCA algorithm to $\{X'_j\}$ and let $\hat{u}$ be the top eigenvector estimate.
4: **Return** $\beta_0 \leftarrow \hat{u}$.

---

**Theorem 4.2** (Linear-time algorithm for spectral initialization). *Consider Definition 2.2. Let $c = \mathrm{ESC}(\beta^\star; f)$, $\delta \in (0, 1)$. Let $C_4$ be hypercontractivity constant of $\tilde{Y} = YX$ as defined in Lemma 4.2. For contamination parameter $\epsilon = O\left(\min\{\frac{1}{C_4^4}, \frac{c^2}{C_4^4(\sigma^2+\mathbb{E}[f^2]+c)^2}\}\right)$, w.p. $\geq 1 - \delta$, the Algorithm 2 takes time $O\left(\frac{md}{C_4^4} \text{polylog}\left(\frac{d}{\epsilon\delta}\right)\right)$ and $m = \Theta\left(C_4^2 \frac{d \log d + \log(1/\delta)}{\epsilon^{3/2}}\right)$ samples to output a unit norm vector $\beta_0$ s.t.*

$$\mathrm{dist}(\beta_0, \beta^\star) = O\left(\frac{C_4 \epsilon^{\frac{1}{4}} \sqrt{\sigma^2 + \mathbb{E}\left[f(X^\top\beta^\star)^2\right] + c}}{\sqrt{c}}\right).$$

*Proof Sketch.* Lemma 4.2 establishes that $\tilde{Y} = YX$ follows a $(4, C_4)$-hypercontractive distribution, where the constant $C_4$ is specified in Lemma 4.2. This property allows us to directly invoke Lemma 2.3, which yields the guarantees for the spectral initialization step of our algorithm. □

*Remark* 4.1. Note that the ESC assumption (Assumption 2.2) alone guarantees the existence of an estimator $\beta_0$ such that $\|\beta_0 - \beta^*\| = O(\epsilon^{1/4})$. Hence, we can efficiently perform robust recovery simply by

### 4.2. Linear Robust Gradient Descent

The second key step, following phase retrieval, is to perform gradient descent on the population risk, $\beta_{t+1} = \beta_t - \eta \nabla \mathcal{L}(\beta_t)$. By Theorem 3.12 of Bubeck (2015) together with Lemma 3.1, the iteration stated above converges linearly to the global minimizer, provided that all iterates remain within the ball $\mathcal{B}(\beta^\star, R)$ and the step size is chosen as $\eta = \frac{2}{\alpha + \gamma}$, where $\alpha$ and $\gamma$ denotes the smoothness and the strong convexity parameters, respectively. Theorem 3.1 implies that, for the population loss $\mathcal{L}(\beta)$, the strong convexity parameter is $\gamma = \frac{\mu}{2}$ and the smoothness parameter is $\alpha = \frac{\mu}{2} + \mu_1$. Since the learning algorithm only has access to the dataset and not the population gradient $\nabla \mathcal{L}(\beta)$, the update stated above cannot be implemented directly. To address this, we follow the standard approach of expressing the gradient as an expectation (Prasad et al., 2020; Buna & Rebeschini, 2025). In particular, $\nabla \mathcal{L}(\beta) = \mathbb{E}\left[(f(X^\top \beta) - Y)f'(X^\top \beta)X\right]$. Substituting the value of $\nabla \mathcal{L}(\beta)$ into above update yields

$$\beta_{t+1} = \beta_t - \eta \, \mathbb{E}\left[(f(X^\top \beta_t) - Y)f'(X^\top \beta_t)X\right].$$

We then replace the expectation with a robust estimator of the gradient. We state the definition of a robust gradient estimator (see Definition 2.1.1 in (Buna & Rebeschini, 2025)) below.

**Definition 4.1.** [Robust Gradient Estimator] Consider a sample $T = \{(x_i, y_i)\}_{i=1}^m$ of size $m$. We call $g(\cdot; T, \delta, \epsilon)$ a gradient estimator if there exist functions $A$ and $B$, where $A, B : \mathbb{N} \times [0,1]^2 \to \mathbb{R}$, such that for any fixed point $\beta \in \mathbb{R}^n$, w.p. $\geq 1 - \delta$,

$$\|g(\beta; T, \delta, \epsilon) - \nabla r(\beta)\| \leq A(m, \delta, \epsilon) \|\beta - \beta^\star\| + B(m, \delta, \epsilon).$$

Following Buna & Rebeschini (2025), we use the notion of a robust gradient estimator, stated formally in Definition 4.1. Let $g_t := g(\beta_t; T, \delta, \epsilon)$ denote such an estimator computed from the dataset $T$. The resulting robust gradient descent update is $\beta_{t+1} = \beta_t - \eta g_t$. We summarize the resulting robust gradient descent procedure in Algorithm 3, and then state its main guarantees.

---

**Algorithm 3** Linear-Robust-Gradient-Descent (LRGD)

**Inputs:** $\beta_0, \delta \in (0,1), \epsilon > 0, P \in \mathbb{N}, \mu, \mu_1$ and datasets $N_2, \ldots, N_{P+1}$.

**Output:** $\beta_P / \|\beta_P\| \in \mathbb{R}^n$

1: Set $\eta = \frac{2}{\mu + \mu_1}$. For $t = 0, \ldots, P - 1$:
  $\triangleright$ Receive contaminated samples $B_t = \{(x_j, y_j)\}_{j=1}^{\tilde{m}}$.
  $\triangleright$ **Gradient Estimation:** For each $(x_j, y_j) \in B_t$, compute $p_j^t = \left(f\left(x_j^\top \beta_t\right) - y_j\right) f'\left(x_j^\top \beta_t\right) x_j$.
  $\triangleright$ Compute $g_t$, the robust mean estimate for $\left\{p_t^j\right\}$ using Robust Mean Estimation (Lemma 2.2).
  $\triangleright$ Update $\beta_{t+1} = \beta_t - \eta g_t$.
2: **Return** $\frac{\beta_P}{\|\beta_P\|}$.

---

**Theorem 4.3.** *Consider $R, \mu$ and $\mu_1$ as defined in Theorem 3.1. Define $\alpha, \gamma, \phi_1$, and $\phi_2$ as in Theorem 4.1. Let $\beta_0 \in \mathcal{B}(\pm \beta^\star, R)$ and contamination parameter*

$$\epsilon = O\left(\min\left\{\frac{\gamma^2}{\phi_1}, \frac{\gamma^2 R^2}{\sigma^2 \phi_2}\right\}\right).$$

*Algorithm 3 takes time $O\left(P\tilde{m}d\log^4\left(\frac{d}{\epsilon\delta}\right)\right)$ and samples $O(P\tilde{m})$ to output an unit norm vector $\beta^{(P)} = \frac{\beta_P}{\|\beta_P\|}$, with probability at least $1 - P\delta$, s.t.,*

$$\left\|\beta^{(P)} - \beta^\star\right\| \leq 2R \exp\left(-P\left(\frac{\gamma}{\alpha + \gamma}\right)\right) + O\left(\frac{\sigma\sqrt{\phi_2 \cdot \epsilon}}{\gamma}\right)$$

*where $\tilde{m} = \tilde{O}\left(d/\epsilon\right)$, and $P = O(1)$ is the number of time-steps in Algorithm 2.*

*Proof Sketch.* We follow a standard inductive argument for gradient descent (Prasad et al., 2020; Buna & Rebeschini, 2025). The objective is to show that the distance (error) to the true signal decreases at each iteration. To establish this, we first derive bounds on the trace and operator norm of the covariance matrix of the gradient of the loss function (Lemma D.2). We then relate the distance to the true signal at the $(t+1)$-th iterate to that at the $t$-th iterate (Lemma D.1). In particular, we show that each iterate satisfies the definition of a robust gradient (Definition 4.1). Finally, we combine these bounds to control the total error across all iterations, as shown in Equation (15) of the paper. $\square$

**Proof of Theorem 4.1.** Under the assumptions on $m$ and $\epsilon$, the output $\beta_0$ of the LRSI algorithm (see Algorithm 2) satisfies $\|\beta_0 - \beta^*\| = O\left(\frac{C_4\left(\sigma^2 + \mathbb{E}[f^2] + c\right)^{1/2}\epsilon^{1/4}}{\sqrt{c}}\right)$. Moreover, under the additional assumption on the corruption level $\epsilon \leq R^4 c^2 / C_4^4\left(\sigma^2 + \mathbb{E}[f^2] + c\right)^2$, we have $\|\beta_0 - \beta^*\| \leq R$. The proof now follows directly from Theorem 4.3.

## 4.3. Applications

In this section, we now demonstrate near-linear time and sample robust recovery for SIMs with 6 different link functions under heavy-tailed noise and strong adversarial contamination. Our representative link functions can be broadly classified into three categories: (i) **Monotonic Links:** Logistic/Sigmoid ($\sigma(z)$), Tanh ($\tanh(z)$), and Probit ($\Phi(z)$). Here, $\phi(z) = \frac{e^{-z^2/2}}{\sqrt{2\pi}}$ and $\sigma(z) = \frac{1}{1+e^{-z}}$, (ii) Phase Retrieval ($f(z) = z^2$), and (iii) **Asymmetric Non-monotonic Links:** GeLU ($z\Phi(z)$), and Swish ($z\sigma(z)$).

**Corollary 4.1.** *Consider the model in Definition 2.2, with the following link functions: Phase Retrieval, GeLU, Swish, Tanh, Probit, and Logistic. There exists an algorithm that, using $n = \tilde{O}(d)$ samples and tolerating a constant fraction $\epsilon$ of adversarial contamination under heavy-tailed noise with variance $\sigma^2$, outputs an estimate $\hat{\beta}$ s.t. $\hat{\beta}$ satisfying $\|\hat{\beta} - \beta^\star\|_2 = O(\sigma\sqrt{\epsilon})$ in $\tilde{O}(nd)$ time, with high probability.*

*Proof.* The proof follows from the fact that of the above link functions satisfy Assumptions 2.1 and 2.2 (see Table 2 in Section E) and Theorem 4.1. □

## 5. Discussion and Future Work

In this work, we established the first framework achieving **near-linear time and optimal sample complexity** for the robust recovery of Single-Index Models with generic, non-monotonic link functions (e.g., GeLU, Swish) under heavy-tailed noise and strong adversarial contamination. Below we outline a few interesting future directions.

**Optimal Error Rates.** Our estimator achieves an $\ell_2$ error rate of $O(\sigma\sqrt{\epsilon})$ under Gaussian covariates in the presence of heavy-tailed noise and $\epsilon$-fraction adversarial contamination. In comparison, for the same setting, Pensia et al. (2025); Cherapanamjeri et al. (2020) established the information-theoretically optimal rate $\tilde{O}(\sigma\epsilon)$. However, for non-linear models with non-convex population loss, existing provably robust algorithms (e.g., phase retrieval), are only known to achieve a $O(\sigma\sqrt{\epsilon})$ rate (Buna & Rebeschini, 2025; Das & Batra, 2026). Our result matches this best-known rate for non-linear single-index models while accommodating a significantly broader class of link functions, including non-monotonic ones. Closing the gap between the achievable rate and the information-theoretically optimal rate for general SIMs remains an important open problem and we leave it for future work.

**Non-Gaussian Covariates.** Our theoretical guarantees heavily leverage the Gaussianity of the design matrix to derive the ESC condition, and in our basin radius analysis by exploiting rotational invariance. We leave extending our proofs to even sub-Gaussian designs as an open question.

**Alternative Loss Landscapes and Adversary Models.** While we focused on the squared loss, investigating if convex basins persist under Huber loss or general $M$-estimators remains an important open question. Additionally, since the existence of a convex-basin does not depend on the corruption model, adapting our framework to Agnostic Learning or Differential Privacy settings is a promising future direction.

**Multi-Index Models (MIMs)** As noted in our introduction, functions like GeLU and Swish are scalar primitives for GLUs, which are inherently Multi-Index Models (MIMs) defined by interactions between multiple projections, as $y = \langle\beta_1, x\rangle \cdot f(\langle\beta_2, x\rangle)$. Extending the guarantees of our work to MIMs (particularly Assumptions 2.1 and 2.2) requires disentangling the interaction terms between multiple weight vectors. This likely necessitates robust tensor decomposition techniques of order significantly higher than those required for SIMs, which presents a distinct set of algebraic and algorithmic challenges.

**Robust Recovery for Links with Information Exponent ($k \geq 3$).** Our framework primarily targets link functions where the signal is detectable via low-order derivatives (specifically, where ESC is non-trivial). However, for link functions with an information exponent $k \geq 3$, the signal is entirely suppressed in lower-order moments. Robustly recovering $\beta^\star$ in this regime would require working with higher-order moment tensors (of order at least $k$). This is an interesting avenue for future work and one concrete line of investigation is discussed next.

**Identifying Label Transforms.** Suppose there exists a map $\tau : \mathbb{R} \to \mathbb{R}$ such that $\tilde{f} = \tau \circ f$ has IE $k^\star \leq 2$ and satisfies Assumptions 2.1 and 2.2. Then our framework applies directly to $\tilde{f}$, yielding efficient robust recovery for the original link function $f$. Identifying such transforms is highly non-trivial, since $k^\star$ is the infimum of the IE over all square-integrable label transforms (Damian et al., 2024, Proposition 2.6), and characterizing which link functions admit a transform $\tau$ such that the resulting $\tilde{f}$ has IE $\leq 2$ and satisfies Assumptions 2.1 and 2.2 is an open problem. We view this as a promising direction for future work, and note that our results provide the first motivation for investigating such regularity conditions on label transforms.

**Empirical Verification.** While our focus is theoretical, empirical evaluation is a vital next step. Implementing high-order robust spectral estimators involves practical engineering challenges, particularly regarding numerical stability and hyperparameter tuning for the filtering subroutines. We leave the extensive experimental benchmarking of these algorithms on real-world datasets, and the potential development of practically optimized heuristics based on our theory, for future work.

## Acknowledgements

The authors are grateful to Ankit Pensia, for many valuable discussions and in particular for pointing us to the robust PCA and robust mean estimation subroutines used in Section 4 which was instrumental in obtaining the near-linear time guarantee of Theorem 4.1. The authors also thank the anonymous reviewers of ICML 2026 for their thorough and constructive feedback, which helped improve the presentation of this work. This work was supported by the Department of Atomic Energy, Government of India, under project no. RTI4014.

## Impact Statement

This paper presents work whose goal is to advance the field of Machine Learning. There are many potential societal consequences of our work, none which we feel must be specifically highlighted here.

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

# A. Literature Review

*Table 1.* Comparative Analysis of Single Index Model Architectures. **Adv. Rob.** stands for Adversarial Robustness. **H.T.** stands for Heavy-Tailed Noise. For both of these columns, ✓ denotes Yes, ✗ denotes No, and ◯ denotes only label corruptions. **Rank** $k$ refers to the first non-zero Hermite coefficient ($k = 1$: Linear, $k = 2$: Quadratic). For **Sample/Time Complexity** columns, $n$ is the sample size, $d$ is the dimension, and $s$ is sparsity.

| Reference | Link Function | Adversarial Robustness | H.T. Noise | Sample Complexity | Time Complexity |
|---|---|---|---|---|---|
| Kalai & Sastry (2009) [Isotron] | Monotone SIM | ✗ | ✗ | $\text{Poly}(d)$ | $\text{Poly}(d)$ |
| Plan et al. (2017) | SIM | ✗ | ✗ | $\tilde{O}(d)$ | $O(d)$. |
| Netrapalli et al. (2015) | Phase Retrieval | ✗ | ✗ | $\tilde{O}(d)$ | $\tilde{O}(nd)$ |
| Candes et al. (2015) | Phase Retrieval | ✗ | ✗ | $\tilde{O}(d)$ | $\tilde{O}(nd)$ |
| Zhang et al. (2018) | Phase Retrieval | ◯ | ✗ | $\tilde{O}(d)$ | $\tilde{O}(nd)$ |
| Lu & Li (2020) | SIM | ✗ | ✗ | $\tilde{O}(d)$ | $O(nd)$ |
| Jagatap & Hegde (2017) | Phase Retrieval | ✗ | ✗ | $\tilde{O}(s^2)$ | $\tilde{O}(ns^2)$ |
| Klivans et al. (2018) | Poly. Regression | ✓ | ✓ | $\text{Poly}(d)$ | $\text{Poly}(d)$ (SoS) |
| Diakonikolas et al. (2019b) | Monotone SIM | ✓ | ✓ | $\tilde{O}(d)$ | $\text{Poly}(d)$ |
| Dong et al. (2025) | Phase Retrieval | ✓ | ✗ | $\tilde{O}(d)$ | $\tilde{O}(nd)$ |
| Diakonikolas et al. (2022) | Logistic Regression | ✓ | ✓ | $\tilde{O}(d^2)$ | $\tilde{O}(nd)$ |
| Awasthi et al. (2022) | Monotone SIM | ✓ | ✗ | $\tilde{O}(d)$ | Heuristic |
| Huang et al. (2026) | Phase Retrieval | ◯ | ✗ | $\tilde{O}(d)$ | $\text{Poly}(d)$ (SDP) |
| Diakonikolas et al. (2019c) | Linear Regression | ✓ | ✗ | $\tilde{O}(d)$ | $\text{Poly}(d)$ |
| Damian et al. (2023) | SIM | ✗ | ✗ | $O(d + d^{k^*/2})$ | $\tilde{O}(nd)$ |
| Buna & Rebeschini (2025) | Phase Retrieval | ✓ | ✓ | $\tilde{O}(d)$ | $\exp^d$ |
| Das & Batra (2026) | Phase Retrieval | ✓ | ✓ | $\tilde{O}(d)$ | $\tilde{O}(n^2d)$ |
| **Our Proposed Framework** | **SIM with positive ESC** | ✓ | ✓ | $\mathbf{\tilde{O}(d)}$ | $\mathbf{\tilde{O}(nd)}$ |

**Monotone link functions without corruption and heavy-tailed noise:** (Kalai & Sastry, 2009) study the recovery of monotone Lipschitz single-index models (SIMs) with an unknown link function $f$ via the Isotron algorithm, which operates in $\text{poly}(d)$ time with $O(1)$ sample complexity. For arbitrary link functions, Plan et al. (2017) provide a linear-time estimator requiring $O(d)$ samples for consistent recovery. Further extending this scope, (Damian et al., 2023) analyze link functions with polynomial-tailed derivatives using online stochastic gradient descent on a smoothed loss function. Their approach achieves a sample complexity of $n = O(d + d^{k^*/2})$ and time complexity of $O(nd)$, where $k^*$ denotes the *information exponent*-the smallest nonzero coefficient of the link function's Hermite expansion under the Gaussian measure.

**Monotone link function under adversarial corruption and heavy-tailed:** Klivans et al. (2018) study linear and polynomial regression with Gaussian covariates, and propose a Sum-of-Squares–based algorithm achieving polynomial sample and computational complexity in $d$. Diakonikolas et al. (2019b) study robust learning in two settings: for SIMs with a monotone (logistic) link, they give a polynomial-time algorithm with sample complexity $\tilde{O}(d)$ achieving generalization error $O(\epsilon^{1/4})$ under logistic loss; for linear regression with heavy-tailed noise, they propose a polynomial-time algorithm with sample complexity $O(d^5)$ achieving estimation error $O(\sqrt{\epsilon})$, where $\epsilon$ denotes the contamination level. (Diakonikolas et al., 2019c) study linear regression under Gaussian noise, and propose a polynomial-time algorithm achieving sample complexity $\tilde{O}(d)$ and estimation error optimal up to constant factors. Awasthi et al. (2022) study monotone SIMs under adversarial contamination and propose a algorithm based on trimmed MLE, achieving sample complexity $\tilde{O}(d)$ with heuristic time complexity.

**Phase retrieval:** Phase retrieval is a classical inverse problem with applications in optics, crystallography, and X-ray imaging. In the non-contaminated, light-tailed setting, the first provable algorithm was due to (Netrapalli et al., 2015), followed by a line of work including (Candès et al., 2015; Jagatap & Hegde, 2017) and others. Netrapalli et al. (2015) propose an alternating minimization approach, while (Candes et al., 2015) use a gradient-based method; both achieve sample complexity $\tilde{O}(d)$ and runtime $\tilde{O}(nd)$. Notably, the method of Netrapalli et al. (2015) requires fresh samples at each iteration, whereas (Candes et al., 2015) does not. Whereas, Jagatap & Hegde (2017) study sparse phase retrieval ($\beta^\star$ is $s$ sparse) and propose an algorithm, achieves sample complexity $\tilde{O}(s^2)$ and computational complexity $\tilde{O}(s^2n)$. Zhang et al. (2018)

study phase retrieval under dense bounded noise and label corruption using a median-truncated Wirtinger Flow algorithm with sample complexity $\widetilde{O}(d)$ and runtime $\widetilde{O}(nd)$, while (Huang et al., 2026) consider same settings as (Zhang et al., 2018) and propose an SDP-based algorithm achieving sample complexity $O(d)$. Phase retrieval under strong adversarial contamination of both labels and covariates has been studied in (Dong et al., 2025; Buna & Rebeschini, 2025; Das & Batra, 2026). Specifically, (Dong et al., 2025) consider the noiseless setting and propose a linear-time algorithm with sample complexity $\widetilde{O}(d)$, while (Buna & Rebeschini, 2025) study heavy-tailed label noise and give an exponential-time algorithm achieving sample complexity $\widetilde{O}(d)$. Subsequently, (Das & Batra, 2026) obtain the same optimal sample complexity $n = \tilde{d}$ with runtime $\widetilde{O}(n^2 d)$.

# B. Details and Omitted Proofs of Section 3

## B.1. Gaussian symmetry

In this section, we state and prove an important property of the Gaussian distribution, which will be used later in the proof of Theorem 3.1.

**Lemma B.1.** *Let* $X \sim \mathcal{N}(0, \mathbf{I}_d)$ *and assume that* $\|\beta^\star\|_2 = 1$. *Let's assume* $g : \mathbb{R} \to \mathbb{R}$ *and* $Z \sim \mathcal{N}(0, 1)$. *Then,*

$$\mathbb{E}[g(X^\top \beta^\star) X X^\top] = \mathbb{E}[g(Z)] \mathbf{I}_d + \left( \mathbb{E}[Z^2 g(Z)] - \mathbb{E}[g(Z)] \right) \beta^\star \beta^{*\top}.$$

*Proof.* Let $X \sim \mathcal{N}(0, \mathbf{I}_d)$ and assume without loss of generality that $\|\beta^\star\|_2 = 1$. Define the scalar random variable $Z := X^\top \beta^\star \sim \mathcal{N}(0, 1)$. By Gaussian orthogonal decomposition, we may write

$$X = Z\beta^\star + U,$$

where $U \perp \beta^\star$, $U \sim \mathcal{N}(0, \mathbf{I}_d - \beta^\star \beta^{*\top})$, and $Z$ and $U$ are independent. Expanding the outer product yields

$$XX^\top = Z^2 \beta^\star \beta^{*\top} + Z(\beta^\star U^\top + U \beta^{*\top}) + UU^\top.$$

Multiplying by $g(Z)$ and taking expectations, the cross terms vanish by independence and symmetry:

$$\mathbb{E}[g(Z) \, Z \, \beta^\star U^\top] = \mathbb{E}[g(Z)Z] \, \mathbb{E}[U] = 0,$$

and similarly for its transpose. Hence,

$$\mathbb{E}[g(Z) \, XX^\top] = \mathbb{E}[g(Z) \, Z^2] \, \beta^\star \beta^{*\top} + \mathbb{E}[g(Z) \, UU^\top].$$

Since $U$ is independent of $Z$ and satisfies $\mathbb{E}[UU^\top] = \mathbf{I}_d - \beta^\star \beta^{*\top}$, we obtain

$$\mathbb{E}[g(Z) \, UU^\top] = \mathbb{E}[g(Z)](\mathbf{I}_d - \beta^\star \beta^{*\top}).$$

Combining terms gives

$$\mathbb{E}[g(Z) \, XX^\top] = \mathbb{E}[g(Z)] \mathbf{I}_d + \left( \mathbb{E}[Z^2 g(Z)] - \mathbb{E}[g(Z)] \right) \beta^\star \beta^{*\top},$$

which establishes the desired decomposition. $\qquad\square$

## B.2. Proof of Theorem 3.1

**Theorem 3.1.** *Let* $Z \sim \mathcal{N}(0, 1)$. *Define the second and fourth moment proxies,* $\mu = \min \left\{ \mathbb{E}\left[f'(Z)^2\right], \, \mathbb{E}\left[Z^2 f'(Z)^2\right] \right\}$, *and* $\mu_1 = \max \left\{ \mathbb{E}\left[f'(Z)^2\right], \, \mathbb{E}\left[Z^2 f'(Z)^2\right] \right\}$. *Consider the model as given in Definition 2.2, s.t. the link function* $f$ *satisfies Assumption 2.1 with radius* $R > 0$, *then for all* $\beta$ *in the Euclidean ball* $\mathcal{B}(\beta^\star, R)$, *the Hessian* $H(\beta) := \nabla^2 \mathcal{L}(\beta)$ *satisfies*

$$\frac{\mu}{2} \mathbf{I}_d \preceq H(\beta) \preceq \left( \frac{\mu}{2} + \mu_1 \right) \mathbf{I}_d.$$

*Proof.* Consider the population loss $\mathcal{L}(\beta) := \frac{1}{2}\mathbb{E}[(f(X^\top \beta) - Y)^2]$ with the corresponding Hessian $\nabla^2 \mathcal{L}(\beta) = \mathbb{E}[(f'(X^\top \beta))^2 + (f(X^\top \beta) - Y)f''(X^\top \beta)) XX^\top]$. The expectation of this Hessian with respect to the noise $\xi$ and covariates $X$ simplifies to:

$$H(\beta) := \mathbb{E}_{X,\xi}[\nabla^2 \mathcal{L}_n(\beta)] = \mathbb{E}_X[Q(X^\top \beta, X^\top \beta^\star) XX^\top]$$

where the function $Q$ is defined as $Q(z, z^*) = (f'(z))^2 + f''(z)(f(z) - f(z^*))$ such that $z = X^\top \beta, z^* = X^\top \beta^\star$. Note that

$$\lambda_{\min}(H(\beta^\star)) = \lambda_{\min}\left(\mathbb{E}_X\left[Q(X^\top\beta^\star, X^\top\beta^\star)XX^\top\right]\right)$$

$$= \lambda_{\min}\left(\mathbb{E}_X\left[(f'(X^\top\beta^\star))^2 XX^\top\right]\right)$$

$$\overset{(a)}{=} \lambda_{\min}\left(\mathbb{E}_{Z\sim\mathcal{N}(0,1)}[(f'(Z)^2]\mathbf{I}_d + (\mathbb{E}_{Z\sim\mathcal{N}(0,1)}[(Z^2f'(Z)^2] - (\mathbb{E}_{Z\sim\mathcal{N}(0,1)}[(f'(Z)^2])\beta^\star\beta^{*\top}\right)$$

$$= \min\{(\mathbb{E}_{Z\sim\mathcal{N}(0,1)}[(f'(Z)^2], (\mathbb{E}_{Z\sim\mathcal{N}(0,1)}[(Z^2f'(Z)^2]\} = \mu,$$

where $(a)$ follows from [Lemma B.1](#) using $f'^2$ as $g$. Similarly,

$$\lambda_{\max}(H(\beta^\star)) = \max\{(\mathbb{E}_{Z\sim\mathcal{N}(0,1)}[(f'(Z)^2], (\mathbb{E}_{Z\sim\mathcal{N}(0,1)}[(Z^2f'(Z)^2]\}.$$

**Convexity and Perturbation Analysis.** To ensure strict convexity, we assume the minimum eigenvalue of the Hessian at the true parameter $\beta^\star$ is bounded by $\mu > 0$. For other $\beta$, we express the Hessian as $H(\beta) = H(\beta^\star) + \Delta(\beta)$, where $\Delta(\beta) = \mathbb{E}[\Delta Q \cdot XX^\top]$ such that $\Delta Q(z, z^*) = (f'(z))^2 - (f'(z^*))^2 + f''(z)(f(z) - f(z^*))$. Using Weyl's inequality, we can say that:

$$\lambda_{\min}(H(\beta)) \geq \mu - \|\Delta(\beta)\|_{op}.$$

**Bounding the Spectral Norm.** We want find an $R$ such that for all $\|\beta - \beta^\star\| \leq R$, $\|\Delta(\beta)\|_{op} \leq \mu/2$, which with Weyl's inequality implies that strong convexity parameter of loss function inside the $R$ radius ball around $\beta^\star$ is $\mu/2$. Now using the Mean Value Theorem, we can say that

$$\Delta Q(z, z^*) = (f'(z))^2 - (f'(z^*))^2 + f''(z)(f(z) - f(z^*))$$

$$\overset{(a)}{=} 3f'(\lambda z + (1-\lambda)z^*)f''(\lambda z + (1-\lambda)z^*)(z - z^*) + f'''(\lambda z + (1-\lambda)z^*)f(\lambda z + (1-\lambda)z^*)(z - z^*)$$

$$= (3f'(\lambda z + (1-\lambda)z^*)f''(\lambda z + (1-\lambda)z^*) + f'''(\lambda z + (1-\lambda)z^*)f(\lambda z + (1-\lambda)z^*))(z - z^*),$$

where $(a)$ follows from Mean Value Theorem and $\lambda, \lambda' \in (0,1)$. Let $Q(z, z^*) = A(z, z^*)(z - z^*)$, where

$$A(z, z^*) := 3f'(\lambda z + (1-\lambda)z^*)f''(\lambda z + (1-\lambda)z^*) + f'''(\lambda z + (1-\lambda)z^*)f(\lambda z + (1-\lambda)z^*).$$

This implies that

$$\|\Delta(\beta)\|_{op} = \sup_{v:\|v\|=1} \mathbb{E}\left[\left|v^\top Q(z, z^*)(z - z^*)XX^\top v\right|\right]$$

$$= \sup_{v:\|v\|=1} \mathbb{E}\left[\left|A(z, z^*)(z - z^*)(X^\top v)^2\right|\right]$$

$$\overset{(a)}{\leq} \sqrt{\mathbb{E}\left[A(z, z^*)^2\right]} \sup_{v:\|v\|=1} \sqrt{\mathbb{E}\left[(z - z^*)^2\right](X^\top v)^4}$$

$$\overset{(b)}{\leq} \sqrt{\mathbb{E}\left[A(z, z^*)^2\right]} \left(\mathbb{E}\left[(z - z^*)^4\right]\right)^{1/4} \sup_{v:\|v\|=1} \left(\mathbb{E}\left[(X^\top v)^8\right]\right)^{1/4}$$

$$\overset{(c)}{=} (315)^{1/4}\sqrt{\mathbb{E}\left[A(z, z^*)^2\right]}\|\beta - \beta^\star\|_2,$$

where $(a)$ and $(b)$ follow from the Cauchy-Schwarz inequality, and $(c)$ follows from the facts that $z - z^* \sim \mathcal{N}(0, \|\beta - \beta^\star\|^2)$ and $X^\top v \sim \mathcal{N}(0, 1)$ when $\|v\| = 1$. Note that $\mathbb{E}\left[A(z, z^*)^2\right]$ depends on path between $\beta$ and $\beta^\star$. So, in order to get an absolute constant, we need to upper bound $\mathbb{E}\left[A(z, z^*)^2\right]$ for every possible path. Now,

$$\mathbb{E}\left[A(z, z^*)^2\right] \overset{(a)}{\leq} 18\mathbb{E}\left[f'(\lambda z + (1-\lambda)z^*)^2 f''(\lambda z + (1-\lambda)z^*)^2\right] + 2\mathbb{E}\left[f'''(\lambda z + (1-\lambda)z^*)^2 f(\lambda z + (1-\lambda)z^*)^2\right]$$

$$\overset{(b)}{=} \mathbb{E}_{Z\sim\mathcal{N}(0,\|\lambda(\beta-\beta^\star)+\beta^\star\|^2)}[18f'(Z)^2 f''(Z)^2 + 2f'''(Z)^2 z^*)^2 f(Z)^2]$$

$$\leq \sup_{\|\beta-\beta^\star\|\leq R, \lambda\in(0,1)} \mathbb{E}_{Z\sim\mathcal{N}(0,\|\lambda(\beta-\beta^\star)+\beta^\star\|^2)}[18f'(Z)^2 f''(Z)^2 + 2f'''(Z)^2 f(Z)^2]$$

$$\overset{(c)}{=} \sup_{\|\beta-\beta^\star\|\leq R} \mathbb{E}_{Z\sim\mathcal{N}(0,\|\beta\|^2)}[18f'(Z)^2 f''(Z)^2 + 2f'''(Z)^2 f(Z)^2],$$

where $(a)$ follows from the inequality $(a+b)^2 \leq 2(a^2+b^2)$, $(b)$ follows from the identity $\lambda z+(1-\lambda)z^* = \lambda(z-z^*)+z^* = X^\top(\lambda(\beta-\beta^\star)+\beta^\star) \sim \mathcal{N}(0, \|\lambda(\beta-\beta^\star)+\beta^\star\|^2)$, and $(c)$ follows from the fact that for $\|\beta-\beta^\star\| \leq R$ and $\lambda \in (0,1)$, the vector $\lambda(\beta-\beta^\star)+\beta^\star$ lies in $\mathcal{B}(\beta^\star, R)$. Now,

$$\sqrt{\mathbb{E}\left[A(z, z^*)^2\right]} \leq \sqrt{\sup_{\|\beta-\beta^\star\| \leq R} \mathbb{E}_{Z \sim \mathcal{N}(0, \|\beta\|^2)}[18f'(Z)^2 f''(Z)^2 + 2f'''(Z)^2 f(Z)^2]}$$

$$\overset{(a)}{=} \sup_{\|\beta-\beta^\star\| \leq R} \sqrt{\mathbb{E}_{Z \sim \mathcal{N}(0, \|\beta\|^2)}[18f'(Z)^2 f''(Z)^2 + 2f'''(Z)^2 f(Z)^2]},$$

where $(a)$ follws from the fact that $(\sup_{x \in A} f(x))^2 = \sup_{x \in A} f(x)^2$, when $f(x)$ is positive over the set $A$ and $\sup_{x \in A} f(x) < \infty$.

Define

$$C_{lip}(R) := \sup_{\|\beta-\beta^\star\| \leq R} \sqrt{\mathbb{E}_{Z \sim \mathcal{N}(0, \|\beta\|^2)}[18f'(Z)^2 f''(Z)^2 + 2f'''(Z)^2 f(Z)^2]}.$$

Note that $C_{lip}(R)$ is finite when $R$ is finite. This implies $\Delta(\beta)$ is a Lipschitz function with the Lipschitz constant being $(315)^{1/4} C_{lip}(R)$ when the space of matrix is endowed with operator norm. Thus, the local convexity condition $\|\Delta(\beta)\|_{op} < \mu/2$ is satisfied when the radius $R$ satisfies:

$$R < \frac{\mu}{2(315)^{1/4} \cdot C_{lip}(R)}.$$

Now,

$$\|H(\beta)\|_{\text{op}} \leq \|H(\beta)\|_{\text{op}} + \|\Delta(\beta)\|_{\text{op}} \leq \lambda_{\max}(H(\beta^\star)) + \mu/2 = \mu_1 + \mu/2.$$

$\square$

## C. Sample splitting in Algorithm 1

*Remark* C.1. Under the notation of Algorithm 1, suppose the dataset consists of $N = C(P+1)d\log(d)$ samples for some absolute constant $C > 0$, among which exactly $K = \varepsilon N$ samples are corrupted. The samples are partitioned uniformly at random, without replacement, into $P+1$ subsets, each containing $Cd\log(d)$ samples. If $d\log(d) \geq C_1 \frac{\log(P+1)+\log(1/\delta)}{\varepsilon^2}$ and $2Cd\log(d) \geq \log(P+1) + \log(1/\delta)$, then with probability at least $1-\delta$, every subset $N_j$, for $j = 1, \ldots, P+1$, contains at most a $2\varepsilon$ fraction of corrupted samples. Thus, without loss of generality, we may assume that after sample splitting in Algorithm 1, each bucket $N_1, N_2, \ldots, N_{P+1}$ contains at most a $2\varepsilon$ fraction of corrupted samples. The same result appears as Claim C.1 in (Das & Batra, 2026), and we therefore omit the proof here, referring the reader to the proof of Claim C.1 in (Das & Batra, 2026) for details.

## D. Omitted Proofs of Section 4

**Lemma 4.1.** *Consider the model given in Definition 2.2 and define $\tilde{Y} := YX$. Then, $\beta^\star$ is the top eigenvector of $\mathbb{E}\left[\tilde{Y}\tilde{Y}^T\right]$ with eigenvalue $\lambda_{\max} = \sigma^2 + \mathbb{E}\left[(f(X^T\beta^\star))^2\right] + 2\mathbb{E}\left[(f'X^T\beta^\star)^2 + f(X^T\beta^\star) \cdot f''(X^T\beta^\star)\right]$.*

*Proof.* Expanding $\mathbb{E}\left[\tilde{Y}\tilde{Y}^T\right]$ we obtain

$$\mathbb{E}\left[\tilde{Y}\tilde{Y}^T\right] = \mathbb{E}\left[(f(X^T\beta^\star) + \zeta)^2 XX^T\right]$$

$$= \mathbb{E}\left[(f(X^T\beta^\star))^2 XX^T\right] + 2\mathbb{E}[\zeta]\mathbb{E}\left[f(X^T\beta^\star)XX^T\right] + \mathbb{E}\left[\zeta^2 XX^T\right]$$

$$= \mathbb{E}\left[(f(X^T\beta^\star))^2 XX^T\right] + \sigma^2 \mathbb{E}\left[XX^T\right].$$

$$= \mathbb{E}\left[(f(X^T\beta^\star))^2 XX^T\right] + \sigma^2 \mathbf{I}_d$$

The second and third equalities follow from linearity of expectation and independence of noise and the bounded second moment of the noise (see Definition 2.2) and the fourth equality follows from distributional assumption that $X \sim \mathcal{N}(0, \mathbf{I}_d)$ (see Definition 2.2). Apply Stein's lemma (Lemma 2.1) on the first term and rearrange to obtain

$$\mathbb{E}\left[\tilde{Y}\tilde{Y}^T\right] = (\sigma^2 + \mathbb{E}\left[(f(X^T\beta^\star))^2\right])\mathbf{I}_d + 2\mathbb{E}\left[(f'(X^T\beta^\star))^2 + f(X^T\beta^\star) \cdot f''(X^T\beta^\star)\right] \beta^\star \beta^{*\top}. \tag{3}$$

The claim now follows from Assumption 2.2. □

**Lemma 4.2.** *Consider the model in Definition 2.2. Then, $\widetilde{Y} = YX$ is $(4, C_4)$ hypercontractive, where*

$$C_4 = 3\left(\mathbb{E}\left[f(X^\top \beta^\star)^8\right]^{1/8} + K_4\right)/\sigma.$$

*Proof.* We have to show that for all $v \in \mathbb{R}^d$, the following holds:

$$\left(\mathbb{E}_{X\sim\mathcal{N}(0,\mathbf{I}_d)}\left[\langle YX, v\rangle^4\right]\right)^{\frac{1}{4}} \leq C_4 \left(\mathbb{E}_{X\sim\mathcal{N}(0,\mathbf{I}_d)}\left[\langle YX, v\rangle^2\right]\right)^{\frac{1}{2}}. \tag{4}$$

Let's consider the left hand side of Equation (4).

$$
\begin{aligned}
\mathbb{E}_{X\sim\mathcal{N}(0,\mathbf{I}_d)}\left[\langle YX, v\rangle^4\right] = \mathbb{E}_{X\sim\mathcal{N}(0,\mathbf{I}_d)}\left[\left(Y\langle X, v\rangle\right)^4\right] &= \mathbb{E}\left[Y^4 \langle X, v\rangle^4\right] \\
&\overset{(a)}{\leq} \mathbb{E}\left[8\left(f(X^\top\beta^\star)^4 + \zeta^4\right)\langle X, v\rangle^4\right] \\
&\overset{(b)}{=} 8\mathbb{E}\left[f(X^\top\beta^\star)^4 \langle X, v\rangle^4\right] + 24K_4^4\|v\|^4 \\
&\overset{(c)}{\leq} 88\sqrt{\mathbb{E}[f(X^\top\beta^\star)^8]}\|v\|^4 + 24K_4^4\|v\|^4 \\
&= \left(88\sqrt{\mathbb{E}[f(X^\top\beta^\star)^8]} + 24K_4^4\right)\|v\|^4,
\end{aligned}
$$

where $(a)$ follows from $(a+b)^4 \leq 8(a^4 + b^4)$, $(b)$ follows from $X^\top v \sim \mathcal{N}(0, \|v\|^2)$ and $\mathbb{E}[\xi^4 \mid X] = K_4^4$, and $(c)$ follows from Cauchy-Schwarz inequality and $X^\top v \sim \mathcal{N}(0, \|v\|^2)$. So, this implies

$$\mathbb{E}_{X\sim\mathcal{N}(0,\mathbf{I}_d)}\left[\langle YX, v\rangle^4\right]^{1/4} \leq \left(88\sqrt{\mathbb{E}[f(X^\top\beta^\star)^8]} + 24K_4^4\right)^{1/4}\|v\| \overset{(a)}{\leq} 4\left(\mathbb{E}[f(X^\top\beta^\star)^8]^{1/8} + K_4\right)\|v\|, \tag{5}$$

where $(a)$ follows from $(a+b)^{1/4} \leq a^{1/4} + b^{1/4}$. Now, let's consider the right hand side of Equation (4).

$$
\begin{aligned}
\mathbb{E}_{X\sim\mathcal{N}(0,\mathbf{I}_d)}\left[\langle YX, v\rangle^2\right] = \mathbb{E}_{X\sim\mathcal{N}(0,\mathbf{I}_d)}\left[\left(Y\langle X, v\rangle\right)^2\right] &= \mathbb{E}\left[Y^2 \langle X, v\rangle^2\right] \\
&\overset{(a)}{=} \mathbb{E}\left[\left(f(X^\top\beta^\star)^2 + \zeta^2\right)\langle X, v\rangle^2\right] \\
&\overset{(b)}{=} \mathbb{E}\left[f(X^\top\beta^\star)^2 \langle X, v\rangle^2\right] + \sigma^2\|v\|^2 \geq \sigma^2\|v\|^2,
\end{aligned}
$$

where $(a)$ follows from $\mathbb{E}[\xi \mid X] = 0$, $(b)$ follows from $\mathbb{E}[\xi^2 \mid X] = \sigma^2$ and $X^\top v \sim \mathcal{N}(0, \|v\|^2)$. So, this implies

$$\left(\mathbb{E}_{X\sim\mathcal{N}(0,\mathbf{I}_d)}\left[\langle YX, v\rangle^2\right]\right)^{\frac{1}{2}} \geq \sigma\|v\|. \tag{6}$$

Note that Equation (4) is satisfied for any positive $C_4$ when $v = 0$. Now,

$$
\begin{aligned}
\sup_{v\mathbb{R}^d:v\neq 0} \frac{\left(\mathbb{E}_{X\sim\mathcal{N}(0,\mathbf{I}_d)}\left[\langle YX, v\rangle^4\right]\right)^{\frac{1}{4}}}{\left(\mathbb{E}_{X\sim\mathcal{N}(0,\mathbf{I}_d)}\left[\langle YX, v\rangle^2\right]\right)^{\frac{1}{2}}} &\overset{(a)}{\leq} \sup_{v\mathbb{R}^d:v\neq 0} \frac{4\left(\mathbb{E}\left[f(X^\top\beta^\star)^8\right]^{1/8} + K_4\right)\|v\|}{\sigma\|v\|} \\
&\leq \frac{4\left(\mathbb{E}\left[f(X^\top\beta^\star)^8\right]^{1/8} + K_4\right)}{\sigma},
\end{aligned}
$$

where $(a)$ follows from Equation (5) and Equation (6). So, the above calculations suggest that one possible choice of $C_4$ is

$$C_4 = \frac{4\left(\mathbb{E}\left[f(X^\top\beta^\star)^8\right]^{1/8} + K_4\right)}{\sigma}.$$

□

**Theorem 4.2** (Linear-time algorithm for spectral initialization). *Consider Definition 2.2. Let $c = \text{ESC}(\beta^\star; f)$, $\delta \in (0,1)$. Let $C_4$ be hypercontractivity constant of $\tilde{Y} = YX$ as defined in Lemma 4.2. For contamination parameter $\epsilon = O\left(\min\{\frac{1}{C_4^4}, \frac{c^2}{C_4^4(\sigma^2 + \mathbb{E}[f^2] + c)^2}\}\right)$, w.p. $\geq 1 - \delta$, the Algorithm 2 takes time $O\left(\frac{md}{C_4^4} \text{polylog}\left(\frac{d}{\epsilon\delta}\right)\right)$ and $m = \Theta\left(C_4^2 \frac{d \log d + \log(1/\delta)}{\epsilon^{3/2}}\right)$ samples to output a unit norm vector $\beta_0$ s.t.*

$$\text{dist}(\beta_0, \beta^\star) = O\left(\frac{C_4\epsilon^{\frac{1}{4}}\sqrt{\sigma^2 + \mathbb{E}\left[f(X^\top\beta^\star)^2\right] + c}}{\sqrt{c}}\right).$$

*Proof.* Lemma 4.1 establishes that for the random vector $\tilde{Y} = YX$, the vector $\beta^\star$ is the leading eigenvector of $\Sigma = \mathbb{E}[\tilde{Y}\tilde{Y}^\top]$. The corresponding largest eigenvalue is

$$\lambda_{\max} = \sigma^2 + \mathbb{E}\left[f(X^\top\beta^\star)^2\right] + 2\mathbb{E}\left[(f'(X^\top\beta^\star))^2 + f(X^\top\beta^\star)f''(X^\top\beta^\star)\right] = \sigma^2 + \mathbb{E}\left[f(X^\top\beta^\star)^2\right] + 2c.$$

Moreover, all remaining eigenvalues are equal, each having value

$$\sigma^2 + \mathbb{E}\left[f(X^\top\beta^\star)^2\right].$$

Lemma 4.2 shows that the random vector $\tilde{Y} = YX$ is $(4, C_4)$-hypercontractive, where

$$C_4 = \frac{4\left(\mathbb{E}\left[f(X^\top\beta^\star)^8\right]^{1/8} + K_4\right)}{\sigma}.$$

We now apply Lemma 2.3 to bound the distance between $\hat{u}$ and the leading eigenvector $\beta^\star$ of $\Sigma$. To do so, we verify that all assumptions of the theorem are satisfied. In the notation of Lemma 2.3, we have

$$\rho = \Theta\left(\frac{16\left(\mathbb{E}\left[f(X^\top\beta^\star)^8\right]^{1/8} + K_4\right)^2}{\sigma^2}\epsilon^{1/2}\right),$$

and

$$\vartheta = \frac{4^6\left(\mathbb{E}\left[f(X^\top\beta^\star)^8\right]^{1/8} + K_4\right)^6}{\sigma^6\epsilon^{1/2}}.$$

To satisfy the sample complexity requirement of Lemma 2.3, we need

$$m = \Theta\left(\vartheta\frac{d \log d + \log(1/\delta)}{\rho^2}\right) = \Theta\left(\frac{16\left(\mathbb{E}\left[f(X^\top\beta^\star)^8\right]^{1/8} + K_4\right)^2}{\sigma^2}\frac{d \log d + \log(1/\delta)}{\epsilon^{3/2}}\right).$$

Furthermore, Lemma 2.3 requires $\rho \leq 1$, which is ensured by the assumption on the contamination level:

$$\epsilon = O\left(\frac{\sigma^4}{\left(\mathbb{E}\left[f(X^\top\beta^\star)^8\right]^{1/8} + K_4\right)^4}\right) \leq 1/2.$$

Thus, all assumptions of Lemma 2.3 are satisfied. Consequently, with probability at least $1 - \delta$, the output $\hat{u} \in \mathbb{R}^d$ of Algorithm $\mathcal{A}_k$ from (Jambulapati et al., 2024) satisfies

$$\|\Sigma\|_{\mathrm{op}} - \mathrm{Tr}\big[\hat{u}^\top \Sigma \hat{u}\big] = O(\rho \|\Sigma\|_{\mathrm{op}}).$$

Recall that the largest eigenvalue of $\Sigma$ is $\sigma^2 + \mathbb{E}[f^2] + 2c$, while all remaining eigenvalues equal $\sigma^2 + \mathbb{E}[f^2]$. Since these two quantities differ, we have $\lambda_1 \neq \lambda_2$, where $\lambda_1$ and $\lambda_2$ denote the largest and second largest eigenvalues of $\Sigma$, respectively. Therefore,

$$\|\Sigma\|_{\mathrm{op}} - \mathrm{Tr}\big[\hat{u}^\top \Sigma \hat{u}\big] \geq \lambda_1 - \big(\lambda_1 \langle \hat{u}, \beta^\star \rangle^2 + (1 - \langle \hat{u}, \beta^\star \rangle^2)\lambda_2\big)$$
$$= (\lambda_1 - \lambda_2)\big(1 - \langle \hat{u}, \beta^\star \rangle^2\big) = 2c\big(1 - \langle \hat{u}, \beta^\star \rangle^2\big).$$

Combining the above bounds yields

$$\big(1 - \langle \hat{u}, \beta^\star \rangle^2\big) = O\left(\rho \frac{\sigma^2 + \mathbb{E}[f^2] + c}{c}\right)$$
$$= O\left(\frac{16\left(\mathbb{E}\big[f(X^\top \beta^\star)^8\big]^{1/8} + K_4\right)^2 (\sigma^2 + \mathbb{E}[f^2] + c)}{\sigma^2 c} \epsilon^{1/2}\right),$$

which further implies that

$$|\langle \hat{u}, \beta^\star \rangle| = \sqrt{1 - O\left(16\left(\mathbb{E}\big[f(X^\top \beta^\star)^8\big]^{1/8} + K_4\right)^2 (\sigma^2 + \mathbb{E}[f^2] + c)\epsilon^{1/2}/\sigma^2 c\right)}.$$

$$\mathrm{dist}(\hat{u}, \beta^\star) = \min\{\|\hat{u} - \beta^\star\|_2, \|\hat{u} + \beta^\star\|_2\} = \sqrt{2 - 2|\langle \hat{u}, \beta^\star \rangle|}$$
$$= \sqrt{2}\sqrt{1 - \sqrt{1 - O\left(\frac{16\left(\mathbb{E}\big[f(X^\top \beta^\star)^8\big]^{1/8} + K_4\right)^2 (\sigma^2 + \mathbb{E}[f^2] + c)\,\epsilon^{1/2}}{\sigma^2 c}\right)}}$$
$$\stackrel{(a)}{\leq} \sqrt{2}\sqrt{1 - \left(1 - O\left(\frac{16\left(\mathbb{E}\big[f(X^\top \beta^\star)^8\big]^{1/8} + K_4\right)^2 \left(\sigma^2 + \mathbb{E}\big[f(X^\top \beta^\star)^2\big] + c\right)\epsilon^{1/2}}{\sigma^2 c}\right)\right)}$$
$$= O\left(\frac{4\left(\mathbb{E}\big[f(X^\top \beta^\star)^8\big]^{1/8} + K_4\right)\left(\sigma^2 + \mathbb{E}\big[f(X^\top \beta^\star)^2\big] + c\right)^{1/2}\epsilon^{1/4}}{\sigma\sqrt{c}}\right)$$
$$= O\left(\frac{C_4\left(\sigma^2 + \mathbb{E}\big[f(X^\top \beta^\star)^2\big] + c\right)^{1/2}\epsilon^{1/4}}{\sqrt{c}}\right),$$

where $(a)$ follow from assumption on the contamination level, namely $C_4^4\big(\sigma^2 + \mathbb{E}\big[f(X^\top \beta^\star)^2\big] + c\big)^2 \epsilon/c^2 = O(1) \leq 1/2$, which implies $C_4^2\big(\sigma^2 + \mathbb{E}\big[f(X^\top \beta^\star)^2\big] + c\big)\epsilon^{1/2}/c = O(1) \leq 1$, together with elementary inequality $\sqrt{1 - x} \geq 1 - x$ when $0 \leq x \leq 1$.

Since we set $\beta_0 = \hat{u}$, it follows from the above that

$$\text{dist}(\beta_0, \beta^\star) = O\left(\frac{C_4\left(\sigma^2 + \mathbb{E}\left[f(X^\top\beta^\star)^2\right] + c\right)^{1/2}}{\sqrt{c}}\epsilon^{1/4}\right).$$

The stated running time follows directly from Lemma 2.3. $\hfill\square$

**Theorem 4.3.** *Consider $R, \mu$ and $\mu_1$ as defined in Theorem 3.1. Define $\alpha$, $\gamma$, $\phi_1$, and $\phi_2$ as in Theorem 4.1. Let $\beta_0 \in \mathcal{B}(\pm\beta^\star, R)$ and contamination parameter*

$$\epsilon = O\left(\min\left\{\frac{\gamma^2}{\phi_1}, \frac{\gamma^2 R^2}{\sigma^2\phi_2}\right\}\right).$$

*Algorithm 3 takes time $O\left(P\tilde{m}d\log^4\left(\frac{d}{\epsilon\delta}\right)\right)$ and samples $O(P\tilde{m})$ to output an unit norm vector $\beta^{(P)} = \frac{\beta_P}{\|\beta_P\|}$, with probability at least $1 - P\delta$, s.t.,*

$$\left\|\beta^{(P)} - \beta^\star\right\| \leq 2R\exp\left(-P\left(\frac{\gamma}{\alpha+\gamma}\right)\right) + O\left(\frac{\sigma\sqrt{\phi_2 \cdot \epsilon}}{\gamma}\right)$$

*where $\tilde{m} = \tilde{O}\left(d/\epsilon\right)$, and $P = O(1)$ is the number of time-steps in Algorithm 2.*

First, we present two key lemmas needed for the proof of Theorem 4.3, followed by the proof itself. The first key lemma expresses the distance of the iterate at time $t + 1$ from $\beta^\star$ in terms of the distance of the iterate at time $t$ from $\beta^\star$. The relation is as follows:

**Lemma D.1.** *Suppose $\beta_t$ obeys $\|\beta_t - \beta^\star\| \leq R$, $\eta \leq 2/(\alpha + \gamma)$ and $g_t$ be the valid robust gradient estimator of population risk at $x_t$ (see Definition 4.1). Then, with probability at least $1 - \delta$, the new iterate $\beta_{t+1}$ obtained according to $\beta_{t+1} = \beta_t - \eta g_t$ satisfies*

$$\|\beta_{t+1} - \beta^\star\| \leq \left(\sqrt{1 - \frac{2\eta\alpha\gamma}{\alpha+\gamma}} + \eta A(m, \delta, \epsilon)\right)\|\beta_t - \beta^\star\| + \eta B(m, \delta, \epsilon) \tag{7}$$

The above lemma and its proof are standard in robust statistics literature; for example, see Theorem 1 in (Prasad et al., 2020) and Lemma 2.2 from (Buna & Rebeschini, 2025). Look at any of the above-mentioned references for the proof of the Lemma D.1. We are omitting the proof here. Our second key lemma computes the trace and operator norm of the covariance matrix of the gradient.

**Lemma D.2.** *Let's assume that $\beta, \beta^\star \in \mathbb{R}^d$ are fixed vectors and $X \sim \mathcal{N}(0, \mathbf{I}_d)$, $y = f(X^\top\beta^\star) + \zeta$. Let $\Sigma = \text{Var}\left((f(X^\top\beta) - y)f'(X^\top\beta)X\right)$ be the variance of the loss gradient. Let's define $\phi_1 := \sup_{\beta\in\mathcal{B}(\beta^\star, R)}(\mathbb{E}\left[f'(X^\top\beta)^{16}\right])^{1/4}$ and $\phi_2 := \sup_{\beta\in\mathcal{B}(\beta^\star, R)}(\mathbb{E}\left[f'(X^\top\beta)^4\right])^{1/2}$. If $\|\beta - \beta^\star\| \leq R$, then*

$$\|\Sigma\|_{\text{op}} \leq 6\phi_1\|\beta - \beta^\star\|^2 + \sqrt{3}\sigma^2\phi_2.$$

*Proof.*

$$\Sigma = \mathbb{E}\left[\left(f(X^\top\beta) - y\right)^2 f'(X^\top\beta)^2 XX^\top\right] - \mathbb{E}\left[\left(f(X^\top\beta) - y\right) f'(X^\top\beta) X\right]\mathbb{E}\left[\left(f(X^\top\beta) - y\right) f'(X^\top\beta) X\right]^\top \tag{8}$$

We now treat each term on the right hand side of Equation (8) separately. Consider the first term $\mathbb{E}\left[\left(f(X^\top\beta) - y\right)^2 f'(X^\top\beta)^2 XX^\top\right]$.

$$\begin{aligned}\mathbb{E}\left[\left(f(X^\top\beta) - y\right)^2 f'(X^\top\beta)^2 XX^\top\right] &= \mathbb{E}\left[\left(f(X^\top\beta) - f(X^\top\beta^\star)\right)^2 f'(X^\top\beta)^2 XX^\top\right] + \mathbb{E}\left[(\zeta)^2 f'(X^\top\beta)^2 XX^\top\right] \\ &\quad - 2\mathbb{E}\left[\left(f(X^\top\beta) - f(X^\top\beta^\star)\right)\zeta\, f'(X^\top\beta) XX^\top\right] \\ &\overset{(a)}{=} \mathbb{E}\left[\left(f(X^\top\beta) - f(X^\top\beta^\star)\right)^2 f'(X^\top\beta)^2 XX^\top\right] + \sigma^2\,\mathbb{E}\left[f'(X^\top\beta)^2 XX^\top\right],\end{aligned}$$

where $(a)$ follows from $\mathbb{E}[\xi \mid X] = 0$ and $\mathbb{E}[\xi^2 \mid X] = \sigma^2$. Now, consider the second term on the right hand side of Equation (8).

$$\mu(f, \beta^\star, \beta) := \mathbb{E}\big[\big(f(X^\top \beta) - y\big)\, f'(X^\top \beta)\, X\big] \overset{(a)}{=} \mathbb{E}\big[\big(f(X^\top \beta) - f(X^\top \beta^\star)\big)\, f'(X^\top \beta)\, X\big],$$

where $(a)$ follows from $\mathbb{E}[\xi \mid X] = 0$. So, this implies

$$\Sigma + \mu(f, \beta^\star, \beta)\mu(f, \beta^\star, \beta)^\top = \Sigma + \mathbb{E}\big[\big(f(X^\top \beta) - f(X^\top \beta^\star)\big) f'(X^\top \beta) X\big]\, \mathbb{E}\big[\big(f(X^\top \beta) - f(X^\top \beta^\star)\big) f'(X^\top \beta) X\big]^\top$$
$$= \mathbb{E}\Big[\big(f(X^\top \beta) - f(X^\top \beta^\star)\big)^2 f'(X^\top \beta)^2\, XX^\top\Big] + \sigma^2\, \mathbb{E}\big[f'(X^\top \beta)^2\, XX^\top\big].$$

As both $\Sigma$ and $\mu(f, \beta^\star, \beta)\mu(f, \beta^\star, \beta)^\top$ are positive definite matrices, hence

$$\Sigma \preceq \mathbb{E}\Big[\big(f(X^\top \beta) - f(X^\top \beta^\star)\big)^2 f'(X^\top \beta)^2\, XX^\top\Big] + \sigma^2\, \mathbb{E}\big[f'(X^\top \beta)^2\, XX^\top\big]. \tag{9}$$

Equation (9) implies

$$\|\Sigma\|_{\mathrm{op}} \leq \left\|\mathbb{E}\Big[\big(f(X^\top \beta) - f(X^\top \beta^\star)\big)^2 f'(X^\top \beta)^2\, XX^\top\Big]\right\|_{\mathrm{op}} + \sigma^2\, \left\|\mathbb{E}\big[f'(X^\top \beta)^2\, XX^\top\big]\right\|_{\mathrm{op}}. \tag{10}$$

We now treat each term on the right hand side of Equation (10) separately. Consider the first term $\left\|\mathbb{E}\Big[\big(f(X^\top \beta) - f(X^\top \beta^\star)\big)^2 f'(X^\top \beta)^2\, XX^\top\Big]\right\|_{\mathrm{op}}$. We can expand it as:

$$\sup_{v:\|v\|=1} v\top \mathbb{E}\Big[\big(f(X^\top \beta) - f(X^\top \beta^\star)\big)^2 f'(X^\top \beta)^2\, XX^\top\Big] v$$

$$= \sup_{v:\|v\|=1} \mathbb{E}\Big[\big(f(X^\top \beta) - f(X^\top \beta^\star)\big)^2 f'(X^\top \beta)^2\, (X^\top v)^2\Big]$$

$$\overset{(a)}{=} \sup_{v:\|v\|=1} \mathbb{E}\Big[\big(f'(\lambda X^\top \beta + (1-\lambda) X^\top \beta^\star)\big)^2 (X^\top(\beta - \beta^\star))^2 f'(X^\top \beta)^2\, (X^\top v)^2\Big]$$

$$\overset{(b)}{\leq} \sqrt{3}\sqrt{\mathbb{E}\Big[\big(f'(\lambda X^\top \beta + (1-\lambda) X^\top \beta^\star)\big)^4 (X^\top(\beta - \beta^\star))^4 f'(X^\top \beta)^4\Big]}$$

$$\overset{(c)}{\leq} \sqrt{3}\left(\mathbb{E}\Big[\big(f'(\lambda X^\top \beta + (1-\lambda) X^\top \beta^\star)\big)^8 f'(X^\top \beta)^8\Big]\right)^{\frac{1}{4}} \mathbb{E}\big[(X^\top(\beta - \beta^\star))^8\big]^{\frac{1}{4}}$$

$$\overset{(d)}{\leq} \sqrt{3}\,\mathbb{E}\Big[\big(f'(\lambda X^\top \beta + (1-\lambda) X^\top \beta^\star)\big)^{16}\Big]^{\frac{1}{8}} \mathbb{E}\big[f'(X^\top \beta)^{16}\big]^{\frac{1}{8}}$$

$$\times \mathbb{E}\big[(X^\top(\beta - \beta^\star))^8\big]^{\frac{1}{4}}$$

$$\leq \sqrt{3}\left(\sup_{\beta \in \mathcal{B}(\beta^\star, R)}\big(\mathbb{E}\big[f'(X^\top \beta)^{16}\big]\big)^{1/8}\right)^2 \mathbb{E}\big[(X^\top(\beta - \beta^\star))^8\big]^{1/4}$$

$$\overset{(e)}{=} \sqrt{3}(105)^{1/4} \sup_{\beta \in \mathcal{B}(\beta^\star, R)} \mathbb{E}\big[f'(X^\top \beta)^{16}\big]^{1/4} \|\beta - \beta^\star\|^2$$

$$\leq 6 \sup_{\beta \in \mathcal{B}(\beta^\star, R)} \mathbb{E}\big[f'(X^\top \beta)^{16}\big]^{1/4} \|\beta - \beta^\star\|^2,$$

where $(a)$ follows from Mean Value Theorem, $(b), (c)$ and $(d)$ follow from Cauchy-Schwarz inequality, $(e)$ follows from fact that $X^\top(\beta - \beta^\star) \sim \mathcal{N}(0, \|\beta - \beta^\star\|_2^2)$ and $(\sup_{x \in A} f(x))^2 = \sup_{x \in A} f(x)^2$, when $f(x)$ is positive over the set $A$ and $\sup_{x \in A} f(x) < \infty$.

Now, consider the second term $\left\|\mathbb{E}\big[f'(X^\top \beta)^2\, XX^\top\big]\right\|_{\mathrm{op}}$.

$$\left\|\mathbb{E}\big[f'(X^\top \beta)^2\, XX^\top\big]\right\|_{\mathrm{op}} = \sup_{v:\|v\|=1} v\top \mathbb{E}\big[f'(X^\top \beta)^2\, XX^\top\big] v = \sup_{v:\|v\|=1} \mathbb{E}\big[f'(X^\top \beta)^2\, (X^\top v)^2\big]$$

$$\overset{(a)}{\leq} \sqrt{3} \sup_{\beta \in \mathcal{B}(\beta^\star, R)} \big(\mathbb{E}\big[f'(X^\top \beta)^4\big]\big)^{1/2},$$

where $(a)$ follows from the Cauchy-Schwarz inequality and the trivial upper bound that for all $\beta \in \mathcal{B}(\beta^\star, R)$, $\left(\mathbb{E}\left[f'(X^\top\beta)^4\right]\right)^{1/2} \leq \sup_{\beta \in \mathcal{B}(\beta^\star, R)} \left(\mathbb{E}\left[f'(X^\top\beta)^4\right]\right)^{1/2}$. Now,

$$\|\Sigma\|_{\mathrm{op}} \leq \left\|\mathbb{E}\left[\left(f(X^\top\beta) - f(X^\top\beta^\star)\right)^2 f'(X^\top\beta)^2 XX^\top\right]\right\| + \left\|\sigma^2 \mathbb{E}\left[f'(X^\top\beta)^2 XX^\top\right]\right\|$$

$$\leq 6 \sup_{\beta \in \mathcal{B}(\beta^\star, R)} \left(\mathbb{E}\left[f'(X^\top\beta)^{16}\right]\right)^{1/4} \|\beta - \beta^\star\|^2 + \sigma^2\sqrt{3} \sup_{\beta \in \mathcal{B}(\beta^\star, R)} \left(\mathbb{E}\left[f'(X^\top\beta)^4\right]\right)^{1/2}$$

$$= 6\phi_1\|\beta - \beta^\star\|^2 + \sigma^2\sqrt{3}\phi_2.$$

$\square$

Now we go for the proof for Theorem 4.3. The proof is an adaptation of the proof of the guarantees of the robust gradient descent algorithm (particularly theorem 3.3) from (Buna & Rebeschini, 2025). The calculations will vary, but the overall flow of arguments remains the same.

*Proof of Theorem 4.3.* Like (Buna & Rebeschini, 2025), first, we try to prove by induction that all iterates $(\beta_t)_{t=0}^{P-1}$ lie inside the ball centered at $\beta^\star$ with radius $R$, because if we can show this then we can write $\|\beta_{t+1} - \beta^\star\|$ in terms of $\|\beta_t - \beta^\star\|$ using Lemma D.1.

**Induction argument**: To avoid redundancy, we consider the case the first case when dist $(\hat{u}, \beta^\star) = \|\hat{u} - \beta^\star\|$. (Otherwise, repeat the same proof with $-\beta^\star$). Note that for the $-\beta^\star$ case, the proof remains the same. The $n = 0$ case follows from the assumptions of the theorem. So, for $n = 0$ case, $\|\hat{u} - \beta^\star\| \leq R$. Let's assume that the induction hypothesis is true till some $t \in \{0, 1, \ldots, P - 1\}, \|\beta_t - \beta^\star\| \leq R$. Now, our goal is to show that $\|\beta_{t+1} - \beta^\star\| \leq R$. Let us recall the definition of a robust estimator of the gradient at the point $x_t$ (see Definition 4.1). It states that $g(\beta_t, T, \delta, \epsilon)$ is a robust gradient estimator of population risk at $\beta_t$ if there exist two functions $A, B : \mathbb{N} \times [0, 1]^2 \to \mathbb{R}$ such that, with probability at least $1 - \delta$, the following bound holds:

$$\|g(\beta_t, T, \delta, \epsilon) - \nabla r(\beta_t)\| \leq A(\tilde{m}, \delta, \epsilon) \cdot \|\beta_t - \beta^\star\| + B(\tilde{m}, \delta, \epsilon).$$

Let's define $g_t := g(\beta_t, T, \delta, \epsilon)$. According to Lemma 2.2, we know that, with probability at least $1 - \delta$,

$$\|g_t - \nabla r(\beta_t)\| = O\left(\sqrt{\|\Sigma\|_{\mathrm{op}} \epsilon}\right), \tag{11}$$

where $\Sigma = \mathrm{Var}\left(\left(f(X^\top\beta) - y\right) f'(X^\top\beta)X\right)$ and $\beta_t$ is treated as a fixed vector. Note that since a fresh sample is used to compute each $g_t$ at each time $t$, we have the following high-probability bound:

$$\mathbb{P}\left(\|g_t - \nabla r(\beta_t)\| = O\left(\sqrt{\|\Sigma\|_{\mathrm{op}} \epsilon}\right) \bigg| \beta_t\right) \geq 1 - \delta. \tag{12}$$

Now, taking expectations with respect to $\beta_t$ on both sides of the above equation 12 gives

$$\mathbb{P}\left(\|g_t - \nabla r(\beta_t)\| = O\left(\sqrt{\|\Sigma\|_{\mathrm{op}}\epsilon}\right)\right) \geq 1 - \delta.$$

Using Lemma D.2, we can say that $\|\Sigma\|_{\mathrm{op}} \leq 6\phi_1\|\beta_t - \beta^\star\|^2 + \sqrt{3}\sigma^2\phi_2$. Now, putting the above bounds and using inequality $\sqrt{a + b} \leq \sqrt{a} + \sqrt{b}$, we get that, with probability at least $1 - \delta$,

$$\|g_t - \nabla r(\beta_t)\| \leq A(\tilde{m}, \delta, \epsilon) \|x_t - x^*\| + B(\tilde{m}, \delta, \epsilon)$$

where $A(\tilde{m}, \delta, \epsilon)$ and $B(\tilde{m}, \delta, \epsilon)$ are defined as

$$A(\bar{m}, \delta, \epsilon) := O\left(\sqrt{\phi_1}\left(\sqrt{\epsilon}\right)\right), \quad \text{and} \quad B(\bar{m}, \delta, \epsilon) := O\left(\sigma\sqrt{\phi_2}\left(\sqrt{\epsilon}\right)\right).$$

So, $g_t$ is a valid robust gradient estimator of population risk at $\beta_t$. So, we have validated all assumptions of Lemma D.1. Using Lemma D.1 we can say that the, with probability at least $1 - \delta$,

$$\|\beta_{t+1} - \beta^\star\| \leq \left( \sqrt{1 - \frac{2\eta\alpha\gamma}{\alpha + \gamma}} + \eta A(\tilde{m}, \delta, \epsilon) \right) \|\beta_t - \beta^\star\| + \eta B(\tilde{m}, \delta, \epsilon), \tag{13}$$

Now, our goal is to show that the right side of Equation (13) is at most $R$. For this, we show that $\sqrt{1 - 2\eta\alpha\gamma/(\alpha + \gamma)} = \frac{\alpha - \gamma}{\alpha + \gamma} < 1, \eta A(\tilde{m}, \delta, \epsilon) \leq \frac{\gamma}{\alpha + \gamma}$, and $\eta B(\tilde{m}, \delta, \epsilon) \leq \frac{\gamma R}{\alpha + \gamma}$ for the chosen values of $\tilde{m}$ and $\epsilon$. Then, using the induction hypothesis $\|\beta_t - \beta^\star\| \leq R$ and the above bounds, we conclude that with probability at least $1 - \delta$, the following holds:

$$\|\beta_{t+1} - \beta^\star\| \leq R. \tag{14}$$

Our induction argument ends here. We now proceed to show one by one that the above bounds hold.

- $\eta A(\bar{m}, \delta, \epsilon) \leq \gamma/\alpha + \gamma$ : Note that,

$$\eta A(\tilde{m}, \delta, \epsilon) \leq \frac{2}{\alpha + \gamma} A(\tilde{m}, \delta, \epsilon) = \frac{\sqrt{\phi_1}}{\alpha + \gamma} O\left( \sqrt{\epsilon} \right).$$

  If $\epsilon$ is chosen to be a small constant, such that $\epsilon \leq \frac{C_2 \gamma^2}{\phi_1}$, then one can adjust the values of $\epsilon$ in such a way that the right-hand side of the above equation is at most $\gamma/(\alpha + \gamma)$.

- $\eta B(\tilde{m}, \delta, \epsilon) \leq \frac{\gamma R}{\alpha + \gamma}$ : Note that,

$$\eta B(\tilde{m}, \delta, \epsilon) = O\left( \frac{\sigma\sqrt{\phi_2}}{\alpha + \gamma} \left( \sqrt{\epsilon} \right) \right) = \frac{\sigma\sqrt{\phi_2}}{\alpha + \gamma} O\left( \sqrt{\epsilon} \right).$$

  As shown previously, if we choose $\epsilon \leq \frac{C_3 \gamma^2 R^2}{\sigma^2 \phi_2}$ then we can adjust the values of $\epsilon$ in a such a way that the right-hand side of the above equation is at most $\frac{\gamma R}{\alpha + \gamma}$.

Note that the above part is similar to the "The induction" section of the proof of Theorem 3.3 from (Buna & Rebeschini, 2025).

**Guarantees for $\beta_P$:** We have shown that with probability at least $1 - P\delta$, Equation (13) and Equation (14) hold for every $t \in \{0, 1, \ldots, P - 1\}$. So, we can conclude that for every $t \in \{0, 1, \ldots, P - 1\}$ :

$$\|\beta_{t+1} - \beta^\star\| \leq \left( \sqrt{1 - \frac{2\eta\alpha\gamma}{\alpha + \gamma}} + \eta A(\tilde{m}, \delta, \epsilon) \right) \|\beta_t - \beta^\star\| + \eta B(\tilde{m}, \delta, \epsilon)$$

$$\leq \left( \frac{\alpha}{\alpha + \gamma} \right) \|\beta_t - \beta^\star\| + \eta B(\tilde{m}, \delta, \epsilon).$$

Iterating this over $t \in \{0, 1, \ldots, P - 1\}$ and using $\|\hat{u} - \beta^\star\| \leq R$, we have

$$\|\beta_P - \beta^\star\| \overset{(a)}{\leq} \left( 1 - \frac{\gamma}{\alpha + \gamma} \right)^P \|\hat{u} - \beta^\star\| + \frac{\eta B(\tilde{m}, \delta, \epsilon)}{1 - \left( \frac{\alpha}{\alpha + \gamma} \right)}$$

$$\leq R \exp\left( -P \frac{\gamma}{\alpha + \gamma} \right) + \frac{B(\tilde{m}, \delta, \epsilon)}{\frac{\gamma}{\alpha + \gamma}},$$

The first and second terms of the right-hand side of step $(a)$ follow from the iteration argument and the sum of the infinite geometric series, respectively. Consider the second term. We know that $\epsilon \leq 1/2$ and this further implies

$$\frac{B(\tilde{m}, \delta, \epsilon)}{\frac{\gamma}{\alpha + \gamma}} = \frac{\frac{\sigma\sqrt{\phi_2}}{\alpha + \gamma} O\left( \sqrt{\epsilon} \right)}{\frac{\gamma}{\alpha + \gamma}} = \frac{\sigma\sqrt{\phi_2}}{\gamma} O\left( \sqrt{\epsilon} \right).$$

By combining the two bounds mentioned above, we obtain

$$\|\beta_P - \beta^\star\| \le R \exp\left(-P\frac{\gamma}{\alpha+\gamma}\right) + \frac{\sigma\sqrt{\phi_2}}{\gamma} O\left(\sqrt{\epsilon}\right). \tag{15}$$

This implies

$$\|\beta_P\| = \|\beta_P - \beta^\star + \beta^\star\| \le \|\beta_P - \beta^\star\| + \|\beta^\star\|$$

$$\le R \exp\left(-P\frac{\gamma}{\alpha+\gamma}\right) + \frac{\sigma\sqrt{\phi_2}}{\gamma} O\left(\sqrt{\epsilon}\right) + 1.$$

Now,

$$\left\|\frac{\beta_P}{\|\beta_P\|} - \beta^\star\right\| = \left\|\frac{\beta_P}{\|\beta_P\|} - \beta_P + \beta_P - \beta^\star\right\| \le \left\|\frac{\beta_P}{\|\beta_P\|} - \beta_P\right\| + \|\beta_P - \beta^\star\|$$

$$= (\|\beta_P\| - 1) + \|\beta_P - \beta^\star\|$$

$$\le 2R \exp\left(-P\frac{\gamma}{\alpha+\gamma}\right) + \frac{2\sigma\sqrt{\phi_2}}{\gamma} O\left(\sqrt{\epsilon}\right).$$

The time complexity of Theorem 4.3 follows from time complexity of Lemma 2.2. $\square$

# E. Numericals

*Table 2.* The explicit values of parameters needed for Theorem 4.1 for different link functions.

| Function | ESC | $\mu$ | $\mu_1$ | $R$ | $C_{lip}(R)$ | $\phi_1$ | $\phi_2$ |
|---|---|---|---|---|---|---|---|
| Logistic/Sigmoid | $3.12 \times 10^{-2}$ | $2.37 \times 10^{-2}$ | $4.48 \times 10^{-2}$ | $3.71 \times 10^{-1}$ | $7.59 \times 10^{-3}$ | $3.27 \times 10^{-3}$ | $5.42 \times 10^{-2}$ |
| Tanh | $1.82 \times 10^{-1}$ | $1.08 \times 10^{-1}$ | $4.64 \times 10^{-1}$ | $4.77 \times 10^{-3}$ | $2.68 \times 10^{0}$ | $6.48 \times 10^{-1}$ | $5.86 \times 10^{-1}$ |
| Probit | $4.59 \times 10^{-2}$ | $3.06 \times 10^{-2}$ | $9.19 \times 10^{-2}$ | $4.73 \times 10^{-2}$ | $7.68 \times 10^{-2}$ | $1.80 \times 10^{-2}$ | $1.09 \times 10^{-1}$ |
| Phase Retrieval | $6.00 \times 10^{0}$ | $4.00 \times 10^{0}$ | $1.20 \times 10^{1}$ | $1.64 \times 10^{-3}$ | $2.89 \times 10^{2}$ | $6.08 \times 10^{2}$ | $6.95 \times 10^{0}$ |
| GeLU | $4.86 \times 10^{-1}$ | $4.56 \times 10^{-1}$ | $5.78 \times 10^{-1}$ | $2.94 \times 10^{-2}$ | $1.84 \times 10^{0}$ | $1.01 \times 10^{0}$ | $6.64 \times 10^{-1}$ |
| Swish | $4.17 \times 10^{-1}$ | $3.79 \times 10^{-1}$ | $5.17 \times 10^{-1}$ | $5.20 \times 10^{-2}$ | $8.67 \times 10^{-1}$ | $7.75 \times 10^{-1}$ | $5.47 \times 10^{-1}$ |

## E.1. Minimum and maximum eigenvalue of the expected Hessian at true signal

From the proof of Theorem 3.1, we know that

$$\mu = \lambda_{\min}(H(\beta^\star)) = \lambda_{\min}\left(\mathbb{E}_x\left[(f'(x^\top\beta^\star))^2 xx^\top\right]\right), \quad \mu_1 = \lambda_{\max}(H(\beta^\star)) = \lambda_{\max}\left(\mathbb{E}_x\left[(f'(x^\top\beta^\star))^2 xx^\top\right]\right)$$

Let $z^* = x^\top\beta^\star$. Since $x \sim \mathcal{N}(0, \mathbf{I}_d)$, then $z^* \sim \mathcal{N}(0, 1)$ because $\|\beta^\star\| = 1$. Due to the rotational symmetry of the Gaussian distribution, the matrix $H(\beta^\star)$ has only two distinct eigenvalues:

1. $\lambda_\|$: Associated with the eigenvector aligned with $\beta^\star$.

2. $\lambda_\perp$: Associated with eigenvectors orthogonal to $\beta^\star$ (with multiplicity $d-1$).

These are computed as:

$$\lambda_\perp = \mathbb{E}_{z^*}\left[(f'(z^*))^2\right]$$

$$\lambda_\| = \mathbb{E}_{z^*}\left[(f'(z^*))^2(z^*)^2\right]$$

We compute these explicitly for each link function. Thus, $\mu = \min\{\lambda_\perp, \lambda_\|\}$, and $\mu_1 = \max\{\lambda_\perp, \lambda_\|\}$.

**E.2. Derivation of $C_{lip}$ and $R$**

The condition for convexity relies on the bound:

$$8.43 \cdot C_{lip} \cdot R < \mu$$

where $C_{lip}$ is defined as:

$$C_{\text{lip}}(R) := \sup_{\|\beta - \beta^\star\| \leq R} \sqrt{\mathbb{E}_{z \sim \mathcal{N}(0, \|\beta\|^2)}[g(z)]},$$

and $g(z) := 18 f'(z)^2 f''(z)^2 + 2 f'''(z)^2 f(z)^2$.

1. **For Bounded Derivatives (Logistic, Tanh, Probit):** From the proof of Theorem 3.1, we have

$$\sqrt{\mathbb{E}[A(z, z^*)^2]} \leq C_{\text{lip}}(R),$$

where

$$A(z, z^*) = 3 f'(z') f''(z') + 2 f'''(z') f(z'),$$

and $z'$ is a point on the line segment joining $z$ and $z^*$. For the case of bounded derivatives, we bound $C_{\text{lip}}(R)$ using a uniform global bound on $|A(z, z^*)|$, namely,

$$C_{\text{lip}}(R) := \sup_{z, z^*} |A(z, z^*)|.$$

Then $R \leq \frac{\mu}{8.43 C_{\text{lip}}(R)}$.

2. **For Unbounded Derivatives (Phase Retrieval, GeLU, Swish, SwiGLU):** For the case of unbounded derivatives, we proceed differently. We first compute a function $g(z)$ such that and then express

$$\mathbb{E}_{z \sim \mathcal{N}(0, \|\beta\|^2)}[g(z)]$$

explicitly as a function of $\|\beta\|$. Substituting $\|\beta\| \leq R + 1$ yields an upper bound of the form $\chi(R)$, where $\chi(\cdot)$ is a deterministic function of $R$.

We then choose $R$ so as to satisfy

$$8.43 \, R \, \chi(R) \leq \mu,$$

using any suitable numerical or analytical method. With this choice, $\chi(R)$ serves as $C_{\text{lip}}(R)$, and the corresponding $R$ defines the radius of the convex basin.

**E.3. Derivations of $c, \phi_1$ and $\phi_2$.**

We know that $c = \text{ESC}(\beta; f) := \mathbb{E}_{Z \sim \mathcal{N}(0,1)}\left[(f'(Z))^2 + f(Z) f''(Z)\right]$. From Theorem 4.3, we know that $\phi_1 = \sup_{\beta \in \mathcal{B}(\beta^\star, R)} \mathbb{E}\left[f'(X^\top \beta)^{16}\right]^{1/4}$ and $\phi_2 = \sup_{\beta \in \mathcal{B}(\beta^\star, R)} \mathbb{E}\left[f'(X^\top \beta)^4\right]^{1/2}$.

1. **For Bounded Derivatives (Logistic, Tanh, Probit):** For the case of bounded derivatives, we obtain a uniform upper bound on $f'$, which can then be used to compute the constants $\phi_1$ and $\phi_2$.

2. **For Unbounded Derivatives (Phase Retrieval, GeLU, Swish, SwiGLU):** Note that

$$\phi_1 = \sup_{\beta \in \mathcal{B}(\beta^\star, R)} \mathbb{E}\left[f'(X^\top \beta)^{16}\right]^{1/4} = \sup_{\beta \in \mathcal{B}(\beta^\star, R)} \mathbb{E}_{Z \sim \mathcal{N}(0,1)}\left[f'(\|\beta\|Z)^{16}\right]^{1/4}$$

$$= \sup_{\|\beta\| \in [1-R, 1+R]} \left(\mathbb{E}\left[f'(\|\beta\|Z)^{16}\right]\right)^{1/4} = \sup_{s \in [1-R, 1+R]} \left(\mathbb{E}\left[f'(sZ)^{16}\right]\right)^{1/4}.$$

Similarly $\phi_2 = \sup_{s \in [1-R, 1+R]} \left(\mathbb{E}\left[f'(sZ)^4\right]\right)^{1/2}$. So, once $R$ is determined, we can use the above forms for $\phi_1$ and $\phi_2$ to compute their values, either via numerical methods or with the aid of standard software packages. In certain cases, such as phase retrieval, these quantities can also be computed in closed form.

## E.4. Code for Numerical Calculations

```python
# -*- coding: utf-8 -*-
"""Calculations
"""

import numpy as np
# Enable JAX 64-bit precision BEFORE importing other JAX modules
from jax import config
config.update("jax_enable_x64", True)

import jax.numpy as jnp
from jax import grad, jit, vmap
from jax.scipy.stats import norm as jax_norm
from scipy.integrate import quad
from scipy.optimize import minimize_scalar, brentq
import pandas as pd

# =========================================
# 1. Definitions of the Functions
# =========================================

def sigmoid(z):
    return 1.0 / (1.0 + jnp.exp(-z))

def probit(z):
    return jax_norm.cdf(z)

def phi_pdf(z):
    return jax_norm.pdf(z)

# Function dictionary mapping names to their JAX implementations
functions = {
    "Logistic/Sigmoid": lambda z: sigmoid(z),
    "Tanh": lambda z: jnp.tanh(z),
    "Probit": lambda z: probit(z),
    "Phase Retrieval": lambda z: z**2,
    "GeLU": lambda z: z * probit(z),
    "Swish": lambda z: z * sigmoid(z),
    "GeGLU": lambda z: z**2 * probit(z),
    "SwiGLU": lambda z: z**2 * sigmoid(z)
}

# =========================================
# 2. Derivative & Expectation Helpers
# =========================================

class FunctionAnalyzer:
    def __init__(self, func_name, func_jax):
        self.name = func_name
        self.f = func_jax

        # JIT compile derivatives for speed
        self.d1 = jit(grad(self.f))
        self.d2 = jit(grad(self.d1))
        self.d3 = jit(grad(self.d2))

    def get_derivatives(self, z):
        """Returns f(z), f'(z), f''(z), f'''(z)"""
        return self.f(z), self.d1(z), self.d2(z), self.d3(z)

    def expected_value(self, integrand_fn, sigma=1.0):
        """
        Computes E[integrand(Z)] where Z ~ N(0, sigma^2).
        We transform variables: Z = sigma * x where x ~ N(0, 1).
        """
        # pdf of standard normal
        norm_pdf = lambda x: (1 / np.sqrt(2 * np.pi)) * np.exp(-0.5 * x**2)

        def wrapper(x):
            z_val = sigma * x
            # Get function terms at z_val
            f, df, d2f, d3f = self.get_derivatives(z_val)
            # Calculate specific integrand term
            val = integrand_fn(z_val, f, df, d2f, d3f)
            # Cast to float to ensure compatibility with scipy.quad
            return float(val * norm_pdf(x))

        # Integrate from -10 to 10.
        # Mass outside [-10, 10] is < 1e-23, which is effectively 0 for integration.
        # points=[0.0] ensures the integrator splits intervals at the peak.
```

```
        res, _ = quad(wrapper, -10.0, 10.0, points=[0.0], limit=100)
        return res

# ========================================
# 3. Calculation Logic for Each Constant
# ========================================

def calculate_all_constants(analyzer):
    print(f"--- Processing: {analyzer.name} ---")

    # --- 0. Calculate ESC (at sigma=1) ---
    # Definition: E[ f'(Z)^2 + f(Z)f''(Z) ]
    def term_ESC_fn(z, f, df, d2f, d3f):
        return df**2 + f * d2f
    val_ESC = analyzer.expected_value(term_ESC_fn, sigma=1.0)

    # --- 1. Calculate mu and mu1 (at sigma=1) ---
    # Term A: E[f'(Z)^2]
    def term_A_fn(z, f, df, d2f, d3f): return df**2
    val_A = analyzer.expected_value(term_A_fn, sigma=1.0)

    # Term B: E[Z^2 * f'(Z)^2]
    def term_B_fn(z, f, df, d2f, d3f): return (z**2) * (df**2)
    val_B = analyzer.expected_value(term_B_fn, sigma=1.0)

    mu = min(val_A, val_B)
    mu1 = max(val_A, val_B)

    # --- 2. Define C_lip(R) Calculator ---
    # C_lip is the sup over beta ball.
    # Ball ||beta - beta*|| <= R implies ||beta|| is in [max(0, 1-R), 1+R].
    # We simplify this to finding max expectation over sigma in this range.

    def get_Clip_integrand_at_sigma(sigma):
        # E[18 f'(Z)^2 f''(Z)^2 + 2 f'''(Z)^2 f(Z)^2]
        def inner(z, f, df, d2f, d3f):
            return 18 * (df**2) * (d2f**2) + 2 * (d3f**2) * (f**2)
        return analyzer.expected_value(inner, sigma=sigma)

    def solve_C_lip(R):
        # We want to maximize the expectation over sigma
        bounds = (max(0.001, 1 - R), 1 + R)

        # minimize_scalar finds minimum, so we minimize negative
        res = minimize_scalar(lambda s: -get_Clip_integrand_at_sigma(s),
                              bounds=bounds, method='bounded')
        return -res.fun # Return the maximum value found

    # --- 3. Solve for R ---
    # Equation: R = mu / (2 * (315^0.25) * C_lip(R))
    # Let g(R) = LHS - RHS. We want g(R) = 0.

    const_factor = 2 * (315**0.25)

    def equation_to_solve(R):
        if R <= 0: return -1.0 # R must be positive
        c_val = solve_C_lip(R)
        # Avoid division by zero if C_lip is 0
        if c_val < 1e-9: c_val = 1e-9
        return R - (mu / (const_factor * c_val))

    # Dynamic Bracketing to find R
    # We check a range. If signs are same, we expand the range.

    low, high = 1e-6, 5.0
    try:
        f_low = equation_to_solve(low)
        f_high = equation_to_solve(high)

        solved_R = 0.01 # Fallback

        if np.sign(f_low) == np.sign(f_high):
            if f_low > 0:
                # Both positive: R > RHS for low and high.
                # This Mean Value Theorems the root is very small (Left of low).
                # Try finding in [1e-9, low]
                try:
                    solved_R = brentq(equation_to_solve, 1e-9, low)
                except:
                    solved_R = 1e-9 # Cap at min
            else:
```

```
                    # Both negative: R < RHS for low and high.
                    # The root is to the right of high.
                    # Try expanding high significantly.
                    try:
                        solved_R = brentq(equation_to_solve, high, 100.0)
                    except ValueError:
                        print(f"Warning: R > 100.0 for {analyzer.name}. Using 100.0")
                        solved_R = 100.0
            else:
                # Signs differ, standard solve
                solved_R = brentq(equation_to_solve, low, high)

    except Exception as e:
        print(f"Error solving R for {analyzer.name}: {e}. Defaulting to 0.01")
        solved_R = 0.01

    # Recalculate C_lip at the solved R for reporting
    final_Clip = solve_C_lip(solved_R)

    # --- 4. Solve for phi_1 and phi_2 ---
    # Both are supremums over the ball (sigma in [1-R, 1+R])

    # phi_1: sup (E[f'(Z)^16])^(1/4)
    def get_phi1_val(sigma):
        def inner(z, f, df, d2f, d3f): return df**16
        return analyzer.expected_value(inner, sigma)

    res_phi1 = minimize_scalar(lambda s: -get_phi1_val(s),
                               bounds=(max(0.001, 1-solved_R),
                               1+solved_R), method='bounded')
    phi1 = (-res_phi1.fun)**0.25

    # phi_2: sup (E[f'(Z)^4])^(1/2)
    def get_phi2_val(sigma):
        def inner(z, f, df, d2f, d3f): return df**4
        return analyzer.expected_value(inner, sigma)

    res_phi2 = minimize_scalar(lambda s: -get_phi2_val(s),
                               bounds=(max(0.001, 1-solved_R),
                               1+solved_R), method='bounded')
    phi2 = (-res_phi2.fun)**0.5

    return {
        "ESC": val_ESC,
        "mu": mu,
        "mu1": mu1,
        "R": solved_R,
        "C_lip(R)": final_Clip,
        "phi1": phi1,
        "phi2": phi2
    }

# ==========================================
# 4. Main Execution
# ==========================================

if __name__ == "__main__":
    results = []

    for name, func_jax in functions.items():
        analyzer = FunctionAnalyzer(name, func_jax)
        try:
            res = calculate_all_constants(analyzer)
            res["Function"] = name
            results.append(res)
        except Exception as e:
            import traceback
            traceback.print_exc()

    # Create DataFrame
    df = pd.DataFrame(results)

    # Reorder columns
    cols = ["Function", "ESC", "mu", "mu1", "R", "C_lip(R)", "phi1", "phi2"]
    if not df.empty:
        df = df[cols]

    print("\n\n=== Final Constants Table ===")
    # Format for nicer reading
    pd.set_option('display.float_format', lambda x: '%.5f' % x)
    print(df.to_string(index=False))
```

