# OpenReview forum: "Convex Basins in Single-Index Model Loss Landscapes: Applications to Robust Recovery under Strong Adversarial Corruption"
_ICML.cc/2026/Conference — ICML 2026 regular_

### Official Review · Reviewer_5GPT · 2026-03-08

**Soundness:** 4
**Presentation:** 3
**Significance:** 3
**Originality:** 3
**Overall Recommendation:** 4
**Confidence:** 4

**Summary:**

The paper leverages developments in robust statistics to design an algorithm for learning the unknown direction of a gaussian single index model with noise in both $x$ and $y$. Specifically, the noise consists of an arbitrary change of a small fraction of examples $(x,y)$ and on top of that a bounded variance $0$ mean noise on $y$. Their algorithm is robust gradient descent with the use of robust PCA for initialization. They are able to solve that problem for single index model link functions that satisfy two conditions: one that ensures that the convexity landscape near the optimal direction is convex and one that ensures that the initialization will fall near the optimal direction. With these conditions and leverage of developments from robust statistics they are able to solve the problem despite a small fraction of strong outliers.

**Compliance With Llm Reviewing Policy:**

Affirmed.

**Final Justification:**

I maintain my final evaluation of weak acceptance. I scored it under for following categories:

Strengths:
1) Soundness: the work proves all its assertions and solve the problem under assumptions.
2) Clarity: the work is well-written and easy to follow.
3) Originality: the work studies a noise model not previously considered to such generality.

Weakness:
1) Significance: the work has certain add hoc assumptions not matching the complexity of the problem and a suboptimal error rate that would likely be surpassed future work.
2) Originality: The problem and assumption choice is original but the techniques use black box theorems from robust statistics and known approaches.

The rebuttal addressed my concerns and helped me understand the context of the work however it reinforced my prior assessment.

**Key Questions For Authors:**

1) The two assumptions seem pretty complex. Can you comment on simpler stronger conditions that imply them?
2) Can you comment on the necessity of the conditions? It seems that no such conditions are present in the agnostic learning literature of the problem. While there is a lot of literature that studies gradient descent dynamics there.

**Limitations:**

Yes

**Strengths And Weaknesses:**

Strengths:
1) Presentation: The paper seems well-written and easy to follow.
2) Soundness: The paper supports all the claims with rigorous proofs.
3) Originality: The paper seems to be one of a few works that handles this problem (specifically with the fraction of arbitrary noise) to this generality.

Weaknesses:
1) Presentation: There are a couple of works with similar settings not cited by the paper. For example:  Learning Geometric Concepts with Nasty Noise, Ilias Diakonikolas, Daniel M. Kane, Alistair Stewart looks to handle the same problem for multi-index models in the classification setting with a similar approach (robustly estimate moments), The Power of Iterative Filtering for Supervised Learning with (Heavy) Contamination Adam R. Klivans, Konstantinos Stavropoulos, Kevin Tian, Arsen Vasilyan works in the same setting for classification however the approach there is more computationally intensive.
2) The paper applies to a wide class of SIM activation functions. However, for a lot of cases the error rate seems sub-optimal since black box robust statistics estimators have been used. For simple cases such as linear regression more complicated techniques have led to better error guarantee.

---

> ### Author Rebuttal · Authors · 2026-03-30
>
> We thank the reviewer for their careful consideration and positive evaluation of our paper. We address the points raised in their review below.
>
> ## Weaknesses
> **On the absence of some related work.** We thank the reviewer for highlighting these relevant works. We will include and discuss these references in the related work section of the revised version of the manuscript.
>
> **On the optimality of the error-rates.** We agree with the observation that information-theoretically optimal error guarantees are achievable for simpler models like linear regression [Pensia et al. 2025], as we have noted in our introduction. Here, the authors first perform covariate filtering, then define a Huber loss (a widely accepted and natural choice for modelling contamination in the response variable), and finally optimize this loss to achieve their desired guarantees.
>
> However, we note that extending these ideas to generic non-linear, non-monotonic SIMs (even for phase retrieval) introduces several significant challenges. One of the main differences is the fact that in the linear regression setting, the Huber loss is convex in the parameter, while for phase retrieval and generic non-linear or non-monotonic SIMs, the loss landscape is inherently non-convex.
>
> This non-convexity fundamentally limits the direct applicability of techniques developed for linear models, even when using robust losses such as the Huber loss, and necessitates new algorithmic and analytical ideas. Nevertheless, we view this as an important and promising direction for future work, and we will explicitly mention both this discussion and the open question in the paper.
>
> ---
>
> ## Questions
>
> **Q1. On Assumptions 2.1 and 2.2.**
>
> ESC (Assumption 2.2) is a higher-order analogue of monotonicity $(\mathbb{E}[f'] > 0)$. In particular, the ESC condition in our paper is actually $\mathrm{ESC}(\beta, f) = \mathbb{E}\left[\left(f^2(X^\top \beta)\right)''\right]$. We use the extended form of ESC since we directly use it in our analysis.
>
> Assumption 2.1 concerns the smoothness of the population objective through expectations of the function $f$ and its derivatives, rather than pointwise smoothness of $f$ itself. In particular, consider the following quantities:
>
> $$
> M_{1} = \sup_{\|\beta - \beta^\star\| \leq R} \left(\mathbb{E}_{Z \sim \mathcal{N}(0,\|\beta\|_2^2)}\left[f(Z)^4\right]\right)^{1/4}, $$
>
> $$
> M_{2} = \sup_{\|\beta - \beta^\star\| \leq R} \left(\mathbb{E}_{Z \sim \mathcal{N}(0,\|\beta\|_2^2)}\left[f'(Z)^4\right]\right)^{1/4},
> $$
>
> $$
> M_{3} = \sup_{\|\beta - \beta^\star\| \leq R} \left(\mathbb{E}_{Z \sim \mathcal{N}(0,\|\beta\|_2^2)}\left[f''(Z)^4\right]\right)^{1/4}, $$
>
> $$
> M_{4} = \sup_{\|\beta - \beta^\star\| \leq R} \left(\mathbb{E}_{Z \sim \mathcal{N}(0,\|\beta\|_2^2)}\left[f'''(Z)^4\right]\right)^{1/4}.
> $$
>
> If $M_1, M_2, M_3, M_4 < \infty$, then Assumption 2.1 is satisfied. In particular, one can show that the local Lipschitz constant satisfies $C_{\mathrm{lip}}(R) \leq 5(M_2 M_3 + M_1 M_4).$ We emphasize that this condition requires boundedness of moments under a Gaussian distribution, which is significantly weaker than assuming uniform boundedness of $f$ and its derivatives everywhere. We now provide some simple scenarios where Assumption 2.1 holds, which demonstrate that Assumption 2.1 holds in a broad class of commonly used settings.
>
> **Bounded functions and derivatives.** If $f, f', f'', f'''$ are uniformly bounded by a constant $M = O(1)$, then clearly $M_1, M_2, M_3, M_4 \leq M$, which immediately implies $C_{\mathrm{lip}}(R) \leq 10M^2$. Hence, the assumption holds.
>
> **Low-degree polynomials.** If $f$ is a low-degree polynomial with coefficients of order $O(1)$, then $f', f'', f'''$ are also polynomials. Since Gaussian moments of any fixed-degree polynomial are finite, it follows that $M_1, M_2, M_3, M_4 < \infty$, and thus $C_{\mathrm{lip}}(R)$ is finite.
>
> We will include this entire exposition in the updated manuscript.
>
> ---
>
> **Q2.**
>
> The identification of necessary conditions for robust learning of general non-linear non-monotonic SIMs under strong adversarial contamination is a major open question in robust statistics. In fact, prior to our work, *not even sufficient conditions were known* for this general setting, and this is precisely the main contribution of our work. We remark here that in the agnostic learning setting we do not assume an explicit *functional* relationship between covariates and responses and instead aim to find a hypothesis whose expected error is close to the optimum within a given benchmark class. This is incomparable with the setting of strong corruption where we assume an explicit functional model relating covariates and measurements (eq.1), where an $\epsilon $-fraction of both covariates and responses can be corrupted. We have already noted that adapting our framework to different adversary models is a promising future direction in our paper.
>
> We thank the reviewer for their positive assessment of this paper.

---

> > ### Author Rebuttal · Reviewer_5GPT · 2026-04-01
> >
> > The authors answers have clarified my questions.  In addition, after reading the other reviewers' comments I have a more clear view of the limitations of the work. I will adjust my score accordingly.

---

> > > ### Author Response · Authors · 2026-04-08
> > >
> > > We would like to clarify a small point in our previous rebuttal on the simplification of Assumption 2.1. The formal statement should be read as follows: If there exists an $R>0$ such that $M_{1}(R),M_{2}(R),M_{3}(R),M_{4}(R)=O(1)$ where $M_i(R)$ is as defined above for all $i\in\\{1,2,3,4\\}$, then there exists a convex basin of radius $\min\\{R,O(1/C_{Lip}(R))\\}$, where $C_{Lip}(R)$ is as defined in the paper.

---

### Official Review · Reviewer_r1rp · 2026-03-08

**Soundness:** 3
**Presentation:** 3
**Significance:** 2
**Originality:** 2
**Overall Recommendation:** 4
**Confidence:** 3

**Summary:**

In this paper, the authors study the problem of robust learning of single-index models in the presence of heavy-tail noises and adversarial corruption of the data.
They show that under certain conditions, the MSE loss has a constant radius convex basin around the ground-truth direction.
By combining this result and existing results on robust initialization and gradient estimation, they obtain an algorithm that robustly learn single-index models with almost-linear sample complexity.

**Compliance With Llm Reviewing Policy:**

Affirmed.

**Final Justification:**

I have raised my score from 3 to 4 as the authors clarified their contribution over existing results. I am not increasing it further, as I am not fully convinced of the significance of the results.

**Key Questions For Authors:**

See the Weakness part for detail. More specifically, I have the following two questions.
* What is the main technical contribution?
* Is it possible to handle the IE > 2 cases by using a different initialization scheme or a suitable label transform?

**Limitations:**

Yes.

**Strengths And Weaknesses:**

Overall, this is an OK paper. It is well-written and easy to follow. The results are also nice.
The issue is that I do not think they are very interesting or surprising.
* The existence of a constant radius convex basin is not surprising, given that it is well-known that even online SGD has a linear convergence rate after weak recovery (and this is true even when the information exponent of the link function is larger than 2). While I don't think this has been explicitly proved before, there do exist local stability results of this style [1-2] and when combined with the linear convergence of SGD after weak recovery, these should have similar effects to the existence of a constant radius convex basin.
* Also, I don't feel the proof of convex basin result is technically interesting, as it is basically Stein's lemma + some perturbation analysis, both of which are somewhat standard tools in this area.
* In addition, the robustness of this algorithm largely comes from existing robust mean estimation and robust PCA algorithms.
* As mentioned by the authors in the discussion section, essentially they require the information exponent (IE) to be at most 2 (I assume this is because they are using existing robust PCA algorithms to escape the initial saddle). It would be more interesting if they can handle single-index models with IE larger than 2. To be fair, after suitable label transforms, almost all link functions have IE at most two [3], but I'm not sure if this type of argument still holds in the setting of this paper.

[1] Yunwei Ren, Eshaan Nichani, Denny Wu, Jason D. Lee. Emergence and scaling laws in SGD learning of shallow neural networks. 2025

[2] Gérard Ben Arous, Murat A. Erdogdu, Nuri Mert Vural, Denny Wu. Learning quadratic neural networks in high dimensions: SGD dynamics and scaling laws. 2025

[3] Alex Damian, Loucas Pillaud-Vivien, Jason D. Lee, Joan Bruna. Computational-Statistical Gaps in Gaussian Single-Index Models. 2024

---

> ### Author Rebuttal · Authors · 2026-03-29
>
> We thank the reviewer for finding our presentation and results nice. We address each comment now.
>
> ## On the Convex Basin
>
> In the clean setting, linear convergence of SGD from any point with non-trivial correlation is behaviorally equivalent to the existence of a convex basin, this is no longer true under strong adversarial contamination. The results in [1] and [2] build on the framework of BAGJ21, who established that in the clean setting, online SGD on SIMs tracks the population gradient flow because the empirical loss landscape faithfully reflects the population loss landscape when data is i.i.d. In [1] (Section 3.3), this framework is extended to MIMs with a fixed number of neurons, and the alignment between each learned direction and the ground truth is tracked as a scalar. The update on each scalar decomposes into a population drift term and a stochastic deviation term. Since each sample is drawn independently of the current iterate, the deviations are mean-zero given the history and form a martingale. Martingale concentration bounds then show the population drift dominates, and convergence follows. In [2] (Section 5.2), a similar argument shows that discrete SGD iterates stay close to the population gradient flow, using operator-norm concentration bounds (Prop. 19, [2]) that rely on Gaussianity of the samples.
>
> However, these proof techniques break down under strong adversarial contamination and heavy-tailed noise since the adversary destroys the mean-zero property of the deviations. Further, in [2], heavy-tailed noise alone breaks the Gaussianity required for (Prop. 19, [2]). The natural approach of filtering out adversarially corrupted samples and then running SGD on the cleaned subset also fails, since any filtering technique produces a data-dependent subset that is no longer i.i.d., breaking concentration arguments. Finally, even with a robust gradient estimator that bounds the deviations, *first-order techniques remain insufficient* to overcome the irreducible bias and guarantee progress toward the signal $\beta^\star$.  We show that strong convexity of the population loss landscape over a constant-radius ball around $\beta^\star$ is sufficient to resolve these issues since the curvature of the loss dominates the adversarial bias (Thm 3.1). We will add this exposition to the revised manuscript.
>
> ---
>
> ## Our Contributions
> 1. We give the first explicit dimension-independent characterization of non-monotonic link functions admitting a strongly convex basin of constant radius in their squared-loss landscape (Assmpn 2.1, Thm 3.1).
> 2. The existence of a convex basin does not by itself yield a computational guarantee: one must also be able to reach the basin efficiently from a random initialization. We identify ESC (Assmp 2.2) as a sufficient computable condition characterizing admissibility into the convex basin.
> 3. We provide the first near-linear time, optimal sample complexity end-to-end algorithm for robust recovery of non-monotonic SIMs (Thm 4.1).
>
> No provable results were previously known for generic non-monotonic SIMs despite robust recovery subroutines. Our contributions are comparable to Cherapanjemeri et al. (NeurIPS 2020) and Diakonikolas et al. (ICML 2022), which also use black-box robust statistics within optimization pipelines. Regarding the reviewer’s comment that our results are uninteresting, we remark that the only related result for non-monotonic SIMs (Awasthi et al., NeurIPS 2022) lacks computational guarantees, whereas we provide provably efficient algorithms.
>
> ## IE > 2
>
> Suppose there exists a map $\tau: \mathbb{R} \to \mathbb{R}$ such that $\tilde{f} = \tau \circ f$ has IE $k^\star\leq 2$ and satisfies Assmps 2.1 and 2.2. Then our framework applies directly to $\tilde{f}$, yielding efficient robust recovery for the original link function $f$. We note that identifying such transforms is a highly non-trivial task; since $k^\star$ is the infimum of the IE $\ell^\star$ over all square-integrable label transforms [Prop 2.6~Dam+24], characterizing which link functions admit a transform $\tau$ s.t. the resulting $\tilde{f}$ has $k^\star\leq 2$ and satisfies Assmps 2.1 and 2.2, is an open problem in its own right, and finding such a label transform immediately yields robust recovery guarantees via our results, underscoring why establishing the IE $\leq 2$ case rigorously is the essential first step.
>
> Hence, our paper also provides motivation for further investigations into the regularity condition on label transforms. E.g., [Lee et al. 2024] construct polynomial label transforms for polynomial SIMs. Extending such constructions to satisfy smoothness and regularity assumptions such as Assmpns 2.1 and 2.2 is an interesting open problem motivated by our work. We will expand Section 5 to make this connection explicit.
>
> ----
>
> We thank the reviewer for their comments, which have allowed us to articulate our contributions more precisely and helped initiate new lines of inquiry.

---

> > ### Author Rebuttal · Reviewer_r1rp · 2026-04-03
> >
> > I thank the authors for the response. It partially addresses my concerns, but I'm still not convinced that the results here are surprising or technically interesting.
> >
> > First, I agree that the usual BAGJ-style online SGD analysis breaks in the presence of an adversary. However, if I understand it correctly, Theorem 3.1 (existence of convex basins) is about the clean population loss. And the fact that the BAGJ-style argument can work is already strong evidence for the existence of the convex basin in the clean population loss. I believe the reason no one has explicitly proved this in the non-adversarial setting is that it is usually cleaner to directly analyze the local behavior around the minimizer, instead of taking the detour via a convex basin argument. Also, as I have mentioned in the initial review, the proof of Theorem 3.1 is basically Stein's lemma + perturbation analysis, and both of them are standard tools. Therefore, I don't find the proof itself to be very technically interesting.
> >
> > Also, I do not feel the contribution of this paper is comparable to [Cherapanamjeri et al., 2020], [Awasthi et al., 2022], or [Diakonikolas et al., 2022]. In the first two papers, the covariates are assumed to follow a general sub-Gaussian distribution, instead of the standard Gaussian distribution. Even in the usual non-adversarial setting, learning single-index model when the input distribution is not spherically symmetric is still mostly open. In [Diakonikolas et al., 2022], the authors consider the streaming model and one of the main contributions of that paper is that streaming algorithm for robust mean estimation. On the other hand, in this submission, the input distribution is the standard Gaussian (and I don't think the proof can be easily extended to general sub-Gaussian distributions as it relies heavily on Stein's lemma) and the robust estimation subroutines are mostly from existing results.

---

> > > ### Author Response · Authors · 2026-04-04
> > >
> > > We thank the reviewer for their continued engagement and address each point in turn.
> > >
> > > We are glad the reviewer agrees that the BAGJ-style analysis breaks under adversarial contamination, which is precisely the setting our paper addresses.
> > >
> > > > "no one has explicitly proved this because it is cleaner to analyze local behavior directly."
> > >
> > > We find this line of reasoning difficult to reconcile with the reviewer's earlier concern that the proof is not technically interesting. The fact that an explicit proof of the existence of a convex basin in our paper is inextricably tied to the **first provable computational guarantees for generic non-monotonic SIMs under strong adversarial contamination and heavy-tailed noise** (a setting in which the local stability style analysis of BAGJ21 and subsequent works breaks entirely) makes a strong case for interestingness.
> > >
> > > > On the proof of Theorem 3.1 being "Stein's lemma + perturbation analysis."
> > >
> > > We would like to stress that standard tools used in combination to establish an interesting result do not make the result uninteresting. The contribution of Theorem 3.1 is not the mechanics of the proof but the result: a dimension-independent strong convexity guarantee over a constant-radius ball for generic non-monotonic link functions.
> > >
> > > ---
> > >
> > > We would like to clarify that the first two listed papers below are examples of fundamental works that *also leverage black-box robust statistical subroutines within known optimization pipelines*. The third listed paper below is the closest work related to ours. We list these to demonstrate that the quantum of our contribution is ICML-worthy.
> > >
> > > 1. Cherapanamjeri et al. 2020 study **linear regression** (which has a well-understood globally convex loss landscape) under adversarial contamination. In contrast, we consider *generic non-monotonic SIMs*. Moreover, the algorithm of Cherapanamjeri et al. 2020 is based on gradient descent, where the gradients are replaced by robust estimates. This robust gradient descent framework builds on prior work by [Prasad et al. 2018], while the construction of the robust gradient estimators further draws on techniques from [Cheng et al. 2019] (see Section 3 of their paper). Their robustified single correction step is also based on a known robust mean estimator applied to the classical one-step estimator in statistics. For these reasons, while our problem setting and technical challenges are fundamentally different, we believe that the quantum of our contributions is directly comparable to theirs.
> > >
> > > 2. We apologize for the earlier comparison with Diakonikolas 2022, which was a typo. We instead wanted to compare our work with Li Cheng Diakonikolas Diakonikolas Ge Wright [NeurIPS 2023]:  “Robust Second-Order Nonconvex Optimization and Its Application to Low Rank Matrix Sensing,” which studies the robust matrix sensing problem under Gaussian designs and Gaussian noise. They employ (see Algorithm A.2) **black-box robust mean estimation** subroutines to estimate both the gradient and the Hessian in a *known non-convex optimization framework Jin et al. ICML 2017*. Hence, the quantum of our contribution is very directly comparable to theirs.
> > >
> > > 3. Finally, Awasthi et al. [NeurIPS 2022] studied generalized linear models under adversarial contamination, with sub-Gaussian covariates, as the reviewer pointed out. However, they consider additive noise with all moments finite, while our setting is more general in terms of noise, as we only require the existence of fourth moments. Moreover, the Trimmed-MLE heuristic was known prior to their work, and their main contribution is a rigorous analysis of this method. In terms of statistical guarantees, they achieve the minimax optimal error rate $\sigma \epsilon$, whereas we achieve an error of order $\sigma\sqrt{\epsilon}$. However, since they used the Trimmed-MLE heuristic as a subroutine, they **lack provable computational guarantees** for their algorithm, whereas we provide a provably linear sample and time complexity algorithm. Hence, again, the quantum of our contribution is directly comparable to theirs.
> > >
> > > ---
> > >
> > > We thank the reviewer for an enriching discussion that allowed us to showcase our contributions in a broader context than what was initially presented in the submission. We will include these detailed expositions in the revised manuscript.

---

### Official Review · Reviewer_yV3b · 2026-03-09

**Soundness:** 3
**Presentation:** 1
**Significance:** 2
**Originality:** 2
**Overall Recommendation:** 4
**Confidence:** 2

**Summary:**

The authors study the problem of learning a single-index model in the presence of heavy-tailed noise and adversarial corruption of the dataset. Their main result identifies a sufficient condition on the link function under which a spectral algorithm can recover the signal in near-linear time. The key insight is that, under this condition, the squared-loss landscape admits a convex basin in a neighborhood of the true parameter. The authors show that a robust spectral method can produce an initial estimator that lies within this basin, after which gradient descent can be used to refine the estimate.

**Compliance With Llm Reviewing Policy:**

Affirmed.

**Final Justification:**

The authors addressed my concerns on the validity of the setting and the relation with other works studying single index models. I am personally not appreciating the way the results are exposed, but the paper is nevertheless understandable. I therefore raised my score

**Key Questions For Authors:**

1. Could you comment on the similarities between your results on the spectral method used at initialization and the one in Mondelli-Montanari? In my reading of that paper the authors focus on phase retrieval as an example, but the results in fact should hold for generic output channel. This should include heavy tail noise and possibly also corrupted observations: one could pick an output channel that with probability $\epsilon$ produces corrupted labels.

2. What are the link functions for which the ESC is not positive?

3. In the conclusion you mention that you would like to compare with link functions with high information exponent. Is this the right quantity to consider? In fact, I would argue that the spectral algorithms in Mondelli-Montanari can succeed in weakly recovering the target even for single index models in the much larger class of generative exponent at most $2$, which in particular include models with arbitrarily high information exponent.

**Limitations:**

Yes

**Strengths And Weaknesses:**

The presentation in the paper is quite sloppy. I think it would increase readability to have a clear statement of what is the problem and what is proven before having all the technical steps explicitly stated. At the current state, Theorem 1.1 appears already in the introduction, but it gives no information on how the learning is done and what are the important aspects that govern the recovery. Looking then into the more technical sections, the algorithms are explained backwards: Algorithm 1 requires knowing already Algorithms 2 and 3 to be parsed. I think the flow would be greatly improved if there was an introduction with just a high level discussion of the results, and then in the more technical sections the learning would be presented in stages: first the spectral part and the conditions on the link functions, and then the rest. Additionally, a number of notational inconsistencies plague the text. The identity matrix is written in all sort of different ways, and it's hard to find a logic for capital and small $x$, $y$: in the first introduction of the model they are the input and output on a single sample and capitalized, then it seems that the single sample quantities should not be capitalized, and that $X$, $Y$ should refer to the stack of the different samples.

Despite these issues in the exposition, this paper appears to be theoretically sound, and the authors offer a very detailed appendix. There are unfortunately no experiments that corroborate the theory, which I feel could be a good addition.

I am not sure of the significance of this problem. Learning single index models is indeed a valid research direction, but I would not be so enthusiastically calling learning single index models "a fundamental problem in high-dimensional statistics". Even accepting the validity of the question, much of the theory of learning single index models has been already studied in abundance in the existing literature referenced by the authors, especially though spectral methods: Mondelli-Montanari 2018 provides spectral methods for a very generic class of single index models. Beyond this, I would question the broader applicability of this work beyond pure theory. In this context, I am not sure how to interpret the remark that the theory applies to "building blocks of modern deep-learning architectures", as the problem is essentially just generalized linear estimation. Studying the fine-tuning though gradient descent is slightly more novel, but the spectral initialization bypasses completely the problem of gaining an initial overlap with the target though gradient descent alone, as one would expect in an actual network.

---

> ### Author Rebuttal · Authors · 2026-03-29
>
> We thank the reviewer and are pleased that they find the paper theoretically sound. We address each concern below.
>
> ---
>
> ## On the Weaknesses
>
> **On the presentation.** We want to note that the overall structure of the paper already follows the reviewer's suggested flow almost exactly.
> - Our introduction already outlines the problem setting and the objective of our work in the first paragraph.
> - In the line above Theorem 1.1, we explicitly state that it is "an informal characterization of the main result". This is intended to give the reader an early understanding of our goal.
> - Similarly, Section 1.1 provides a high-level technical overview before any formal development.
> - The main formal result (Theorem 4.1) and the corresponding algorithm (Algorithm 1) are presented together in Section 4.
>
> Algorithm 1 is presented top-down as a composition of two subroutines, LRSI (Algo. 2) and LRGD (Algo. 3), following immediately after the theorem that motivates it. We note that this top-down modular structure is standard in the algorithmic robust statistics literature; c.f. Diakonikolas et al. [ICML 2022] and Awasthi et al. [NeurIPS 2022], which adopt the same convention.
>
>
> > "...notational inconsistencies..."
>
> We apologise for the oversight regarding notational inconsistencies and will correct them in an updated version of the manuscript. We clarify that we intended to use capital letters for random variables and lowercase letters for their realizations, as stated in the notation paragraph of Section 2.
>
> **On SIMs.** Historically, SIMs have unified linear regression, logistic regression, phase retrieval, and generalized linear models within a single framework [Box-Cox 1964, Ichimura 1993], and have recently become the canonical testbed for computational-statistical gaps in non-linear inference [Ben Arous et al. 2021, Damian et al. 2024, Joshi et al. 2025], and for studying dynamics in neural networks [Ben Arous-Hsu 2025]. Since SIMs capture non-linear relationships through a low-dimensional projection and thus provide a tractable framework for analyzing key phenomena in more complex architectures such as neural networks, we believe that the robust learnability of SIMs is an important question.
>
> **On connection to Deep Learning.** Our paper provides the next rung in achieving (provably) robust machine learning from the perspective of robust statistics for SIMs. Modern ML architectures utilize link functions like GeGLU and SwiGLU, for which no results were known for strong adversarial corruptions prior to our work. Furthermore, historically, guarantees for two-stage algorithms have been followed by guarantees for pure gradient descent (e.g., for phase retrieval, [Netrapalli et al. 2013], [Candes et al. 2015]'s provable two-stage algorithms were followed by analysis of gradient descent from random initialization, [Chen et al. 2018]); hence, we anticipate that our work too will likely be followed by analysis of gradient descent alone for such problems, which is insightful for contemporary robust ML (as already noted by the reviewer).
>
> **On Experiments.** We agree that empirical validation is valuable and note it as future work. However, theory-only contributions are standard in this area; for instance, Awasthi et al. (NeurIPS 2022) provide purely theoretical results without experiments.
>
> ---
>
> ## Questions
>
> **Q1. Differences with MM18:**
> - **Asymptotic vs finite-sample:** MM18 gives asymptotic learnability via the spectral threshold $\delta_u$ (as $n/d \to \infty$), with no finite-sample guarantees. We provide explicit finite-sample guarantees: linear sample complexity and near-linear runtime. ESC is a closed-form condition on $f$ enabling such finite-sample estimation.
> - **Noise model:** MM18 considers stochastic, i.i.d., label-only noise (independent across samples). This is formally incomparable to our setting (Definition 2.1), where an adaptive adversary can arbitrarily corrupt up to $\varepsilon n$ samples, affecting both covariates and responses.
>
> **Q2.** A concrete example with $\text{ESC} < 0$ is $f(z) = e^{-z^2}$, giving $\text{ESC} = -2/5^{3/2} < 0$. This function has no natural interpretation as an activation function, and we are not aware of any practically relevant link function for which $\text{ESC} \leq 0$
>
> **Q3.** Our framework also applies to SIMs $f$ which after composing with a proper label transform $\mathcal{T}$ have IE$\leq 2$ (i.e., GE $\leq 2$ for $\mathcal{T}\circ f$) satisfies Assmpns 2.1 and 2.2. Unlike the asymptotic regime guarantees of MM18, we also give linear sample and time robust recovery guarantees for generic non-monotonic SIMs. We note here that identifying transforms that reduce GE while preserving our smoothness conditions and near-linear guarantees under adversarial contamination is an interesting open problem.
>
> We will update Section 5 accordingly.
>
> We again thank the reviewer for their positive assessment of the paper's technical soundness.

---

> > ### Author Rebuttal · Reviewer_yV3b · 2026-04-03
> >
> > I thank the authors for their rebuttal, and I will update my score.
> >
> > I agree that SIM are interesting objects to study theoretically, but I would not necessarily say that they have an immediate implication in the study of actual deep networks. This is also in agreement with the rebuttal, where I do not see anything about the impact on the study of more realistic architectures. I would change the wording in the main text to reflect this.

---

> > > ### Author Response · Authors · 2026-04-04
> > >
> > > We agree with the reviewer that our paper does not study transformer architectures or multi-layer networks. The broader architectural context in the paper is intended as motivation for why these specific link functions are worth studying, not as a claim of direct applicability to full networks. Extending our results to Multi-Index Models, which directly encompass GeGLU and SwiGLU, is already explicitly identified as a future direction in Section 5. We will make this phrasing more explicit in the introduction to clarify any misunderstandings. We hope that this addresses all remaining concerns of the reviewer.

---

### Official Review · Reviewer_mX7r · 2026-03-12

**Soundness:** 3
**Presentation:** 4
**Significance:** 3
**Originality:** 2
**Overall Recommendation:** 4
**Confidence:** 3

**Summary:**

This paper studies robust recovery in Gaussian single-index models of the form $Y = f(X^\top \beta^\ast) + \zeta$, under the simultaneous presence of heavy-tailed noise and a constant fraction of strong adversarial contamination. The main goal is to understand when one can efficiently recover the unknown direction $\beta^\ast$ for a broader class of nonlinear, potentially non-monotone link functions than those covered by prior work. To this end, the paper introduces two main structural conditions. First, Assumption 2.1 is used to establish that the population squared loss admits a dimension-independent constant-radius convex basin around the ground truth. Second, Assumption 2.2, termed Expected Squared Convexity (ESC), is used to guarantee that a moment-based spectral initialization aligns with the signal direction and enters this basin even under contamination. Building on these two ingredients, the paper combines robust spectral initialization with robust gradient descent, improving the initialization error from $O(\epsilon^{1/4})$ to a final estimation error of order $O(\sigma \sqrt{\epsilon})$, while claiming near-linear sample complexity $\widetilde{O}(d)$ and running time $\widetilde{O}(nd)$. The framework is further instantiated for several representative link functions, including Logistic, Tanh, Probit, Phase Retrieval, GeLU, and Swish, with supporting parameter tables provided in the appendix.

**Compliance With Llm Reviewing Policy:**

Affirmed.

**Final Justification:**

My main concerns have mostly been resolved, so I will keep my current score unchanged.

**Key Questions For Authors:**

1. All theoretical results are limited to Gaussian covariates. Can you provide partial theoretical guarantees or extended analysis for sub-Gaussian data which is more common in practice?

2. The estimation error
$O(σ
ϵ
 )$
 is worse than the information-theoretic optimal value. What is the essential bottleneck for non-monotonic link functions?

3. The hidden polylogarithmic factors in time complexity are not quantified. How do they affect the actual running efficiency in high-dimensional scenarios?

**Limitations:**

Yes.

**Strengths And Weaknesses:**

Strengths:

Soundness: The theoretical derivation is rigorous, the assumptions are clearly defined, and the proofs of convex basin, ESC condition and algorithm guarantees are complete and reasonable under Gaussian design.

Originality: It breaks through the limitation of previous research that only focuses on monotonic link functions and phase retrieval, and extends the robust recovery theory to the core activation functions of deep learning.

Significance: It builds a bridge between high-dimensional robust statistics and modern deep learning architectures, providing theoretical support for the robust learning of Transformer-related models.

Presentation: The paper is well-structured, with clear technical overview and detailed numerical verification for six typical link functions.

Weaknesses:

Soundness: All theoretical guarantees strictly rely on Gaussian covariates, without any extension to sub-Gaussian or non-Gaussian data, which greatly reduces the practical application value.

Originality: While the overall framework successfully extends classical phase retrieval techniques to a much richer family of models, the overall two-stage pipeline (spectral initialization + gradient descent) follows a well-established paradigm in non-convex robust estimation.

Significance: There is no empirical experiment on synthetic or real data, and the actual performance of the algorithm such as convergence speed and anti-pollution ability is not verified.

Presentation: The appendix proofs are excessively simplified, lacking intuitive explanation of key steps, which makes it difficult for readers to reproduce the results.

---

> ### Author Rebuttal · Authors · 2026-03-30
>
> We thank the reviewer for their careful review of our paper. We address the specific concerns below.
>
> ## Weaknesses
> 1. **Beyond Gaussian Covariates.** A useful historical pattern contextualizes our choice. As we note in the first paragraph of our paper, foundational algorithmic results for non-linear SIMs, including the seminal works on phase retrieval [Netrapalli et al. 2013, Candès et al. 2015], spectral initialization [Lu-Li 2020, Mondelli-Montanari 2018], and general SIM recovery [Damian et al. 2024], were all first established under Gaussian covariates and only later extended to sub-Gaussian/non-Gaussian settings. We therefore view our results as a foundational step in robust SIM learning. Extending our techniques to more general covariate distributions, such as sub-Gaussian or other non-Gaussian models, is an important and natural direction for future work, as we have already noted explicitly in Section 5.
> 2.  **On the two-stage pipeline.** We agree with the reviewer that the two-stage pipeline is a well-established paradigm in non-convex robust estimation problems. However, before our work, it was not clear whether or why such a framework should extend to generic non-monotonic link functions, especially under strong adversarial contamination, and yield nearly-optimal computational guarantees. Our contribution is precisely identifying the specific properties (see Assumptions 2.1 and 2.2) that allow us to establish that the two-stage pipeline extends to generic non-monotonic SIMs.
> 3. **On empirical validation** We agree that empirical validation is a valuable next step and have already noted it as future work in the paper. We note that it is standard in this area of research to produce theory-only contributions. For example, the most directly comparable prior work in our setting, Awasthi et al. [NeurIPS 2022], presents theoretical results without empirical evaluation.
> 4. **On proof sketches.** We have already outlined most of the proof sketches for Theorem 3.1 and Theorem 4.2 following the main statements. For Theorem 4.2, the paragraph in Section 4.1 preceding the theorem already provides intuition and a high-level understanding of the proof. For Theorem 4.3, we acknowledge that the provided proof sketch is quite brief. We will include an extended proof sketch in the updated manuscript.
> > We follow a standard inductive argument for gradient descent. The objective is to show that the distance (error) to the true signal decreases at each iteration. To establish this, we first use bounds on the trace and operator norm of the covariance matrix of the gradient of the loss function (Lemma D.2). We then relate the distance to the true signal at the $(t+1)$-th iterate to that at the $(t)$-th iterate (Lemma D.1). In particular, we show that each iterate satisfies the definition of a robust gradient (Definition 4.1). Finally, we aggregate these bounds to control the total error across all iterations as shown in Equation 19 of the paper.
>
>
> ## Questions
> 1. We thank the reviewer for this question. We indeed acknowledge that currently, we do not have a clear approach to extend our analysis to the sub-Gaussian setting. The main technical challenge arises in the spectral initialization step, which crucially relies on the Gaussian structure. In particular, under Gaussian covariates, the top eigenvector of the covariance matrix of $(Y X)$ aligns with the true signal $(\beta^\star)$ (see Lemma 4.1), which forms the basis of our spectral initialization procedure. For sub-Gaussian covariates, this property does not necessarily hold, and therefore, the current proof techniques break down at this stage. Addressing this issue would likely require new ideas for initialization under weaker distributional assumptions.
> We have highlighted this limitation and explicitly mentioned it as an open problem in the discussion and future work section of our paper.
> 2.  Once again, we thank the reviewer for this question. We have already acknowledged the gap between our current estimation error, which is of the order $O(\sigma \sqrt{\epsilon})$, and the information-theoretic optimal rate is $O(\sigma \epsilon)$ in the paragraph titled “Discussion on the error rate” of our paper. We have also listed it as an important open problem in the paper.
> 3. We will make the polylogarithmic factors in our analysis more explicit in the updated manuscript. For both the spectral initialization procedure (Theorem 4.2) and the gradient descent algorithm (Theorem 4.3), the polylogarithmic factor is of the order $O\left(\log^4\left(\frac{d}{\epsilon \delta}\right)\right)$. Overall, these factors grow only polylogarithmically with the dimension ($d$), and hence have a mild impact on the running time even in high-dimensional regimes.
>
> We once again thank the reviewer for their positive evaluation of our paper.

---

> > ### Author Rebuttal · Reviewer_mX7r · 2026-04-02
> >
> > I sincerely thank the authors for their rebuttal that addressed my concerns.

---

### Decision · Program_Chairs · 2026-04-30

**Decision:**

Accept (regular)

**Comment:**

The submission analyzes robust recovery of single index models.  The informed reviewers all agree the contribution is valuable and I thus recommend acceptance.